# Sex differences in avian parental care patterns vary across the breeding cycle

Daiping Wang [1,2] ✉, Wenyuan Zhang [3,4], Shuai Yang[1,2] & Xiang-Yi Li Richter [5] ✉

Parental care in birds consists of elaborate forms across stages, including nest building, incubation, and offspring provision. Despite their evolutionary importance, knowledge gaps exist in the extent to which parents contribute disproportionately to these forms and factors that are associated with variations in care patterns between sexes. Here, we analyzed 1533 bird species and discovered remarkable variability in care patterns. We show that parental care should not be viewed as a unitary trait but rather as a set of integrated features that exhibit considerable temporal and sex-specific variation. Our analyses also reveal moderate consistency in care patterns between breeding stages, pointing towards shared intrinsic factors driving sex-specific care. Notably, we found that species experiencing strong sexual selection on males or species facing paternity uncertainty display a tendency towards female-biased care. This work advances our understanding of the temporal variations in sex-specific contributions to avian parental care and their potential evolutionary drivers.

Birds often provide extensive parental care that enhances their offspring's survival and future reproductive fitness. Avian parental care comprises diverse forms, including nest building, incubation, provisioning the offspring, and defending them against predators[1–3]. Despite the benefits that parental care brings, it also brings costs such as energy, time, the opportunity for extra-pair mating, and/or the potential for starting a new clutch. Furthermore, it may increase the predation risk of the parents. Consequently, there are conflicts between the male and female parent and between parents and offspring. In cooperative breeding species, the conflicts further involve helpers of different relatedness with the breeders and the dependent offspring. These intricate relationships have inspired theoretical studies about the optimal parental care strategies of each parental sex and helpers. Early models characterized parental care as an all-or-nothing choice between deserting and caring[4–7], while later models generally treated parental investment as a continuous trait[8,9]. Most theoretical work, however, treats parental care as a unitary trait rather than a

composite of several functionally integrated characteristics[10,11]. A few rare exceptions have considered task specialization between parents, such as feeding the young and defending them from predators[8,12], but these models do not predict how parents contribute to different tasks over time across a breeding cycle.

Focusing on one or a small set of species, optimal levels of parental efforts have been studied as functions of various factors, including brood quality[13,14], the certainty of paternity[15–17], operational sex ratio and sexual selection[18,19], and sex-specific life history characters such as adult mortality[20] and the ability to care[11]. Special attention has been paid to which sex should provide care, yet without distinguishing care forms across the whole breeding cycle[21–24]. It is not clear if sex-specific parental strategies differ across distinct care forms. In other words, if one sex has participated in nest building, would it also incubate the eggs laid in that nest and/or feed the chicks after they hatch? In birds, the variations, temporal consistency, and potential drivers of sex differentiation in providing care across different care

[1]Key Laboratory of Animal Ecology and Conservation Biology, Institute of Zoology, Chinese Academy of Sciences, 1 Beichen West Road, Chaoyang 100101 Beijing, China. [2]University of Chinese Academy of Sciences, 100049 Beijing, China. [3]Edward Grey Institute, Department of Biology, University of Oxford, Oxford OX1 3SZ, UK. [4]Department of Biology, Quebec Centre for Biodiversity Science, McGill University, Montreal, QC H3A 1B1, Canada. [5]Institute of Ecology and Evolution, University of Bern, Baltzerstrasse 6, CH-3012 Bern, Switzerland. ✉e-mail: wangdaiping@ioz.ac.cn; li@evolbio.mpg.de

stages remain largely unknown. To assess how much the contributions of the male and female parent differ between distinct care forms across a breeding cycle, we first survey the participation of each parent in three typical care forms (i.e., nest building, incubation, and offspring provisioning) across broad avian taxa. We then explore the temporal consistency of sex-difference in contribution between consecutive stages of parental care. We ask: if a male has built the nest, would he continue to incubate the eggs, or leave the task to the female? Similarly, if a female has incubated the eggs, would she continue to feed the chicks after they hatch, or leave them to the male?

Here, we statistically test whether the temporal consistency of parental care follows one of the three patterns, namely, being consistent (positive correlation between stages), complementary (negative correlation between stages), or irregular (no significant correlation between stages). On the one hand, increasing evidence for consistent individual behavior across taxa and social contexts suggests that behavioral consistency may be beneficial[25–29]. For example, being consistently bolder or more active than others may benefit the growth and fecundity of the focal individual under certain conditions[30]. In the case of parental care, it may be favorable for one sex to specialize and consistently provide care across different forms. On the other hand, considering the conflicts between parents over costly caring efforts, flexible behavioral responses might be more adaptive[31]. Indeed, theoretical[32] and empirical studies found that males and females can communicate and negotiate their parental effort[33–37], and the negotiation rules can be sex-specific[38]. Therefore, it is possible that parents negotiate with each other and take turns providing care across consecutive breeding stages. For instance, at the stage of nest building, if the male has built the nest solely, the female may agree to provide care alone in the next stage (i.e., incubation), followed by the male joining her in offspring provision thereafter. Furthermore, it is also possible that parental efforts of males and females are neither consistent nor complementary, but irregular, determined by idiosyncratic factors such as sex-specific opportunity cost associated with different forms of care. For example, caretaking males of the black coucals were found to have different success rates in siring extra-pair offspring at varying stages of parental care[39], suggesting that the opportunity cost of parental care can change over time. Given the evidence pointing toward all three possibilities, we build statistical models to find the general temporal consistency pattern of parental care across avian taxa, which may aid future studies that aim to uncover the causal pathways leading to the stage-specific parental care pattern in different bird species.

Besides testing the temporal consistency of parental care, we also aim to identify the driving forces of the variations of sex-difference in parental contributions across different care forms. We test the roles of sexual selection, certainty of paternity, nest predation risk, and offspring's life history traits in driving the variations. The four potential driving forces are chosen because there are clear theoretical predictions of their effects on sex differences in parental care[40]. Strong sexual selection on males is predicted to produce female-biased care[18,19], except when females prefer to mate with care-providing males, which can lead to the evolution of male-biased care[41]. Sexual selection has also been shown to associate with evolutionary transitions between major patterns of parental care[42,43]. Another important predictor of parental care investment is the certainty of parentage. Although intuition suggests that the difference between male and female parents in expected parentage (e.g., due to female extra-pair mating) should produce female-biased care, theoretical models showed that the impact of paternity on care could be positive, negative, or non-significant[15–17,44]. In addition, sex differences in parental care can arise if providing care is more costly or less efficient for one sex than the other. For example, high nest predation risk may select for female-biased care when females have more cryptic plumages than males, which is common in passerines[45]. Being drabber, females may be less likely to attract predators to the offspring and themselves when

providing care. Furthermore, the life history traits of offspring are of interest to study because they reflect broods' reproductive value and needs. Because parents' caring efforts are linked to the trade-off between their current and future reproductive fitness, they are often expected to invest more in broods of higher reproductive value, e.g., broods of (artificially) enlarged sizes[13,14,46–48]. However, theoretical models and experiments showed that parental investment may not always increase with the brood's reproductive value, with the adaptive behavior sensitively dependent on environmental factors such as food supply[49,50].

We collected parental care data from all 1533 avian species in the Birds of the World database[51], tested the consistency of sex-difference in care patterns across nest building, incubation, and offspring provision, and identified the main driving forces of variations in the care pattern at different stages of a breeding cycle.

We found significant sex-difference in parental contributions across the three forms of parental care. The temporal consistency of care patterns between breeding stages was moderate. Importantly, species subject to intense sexual selection on males, or those with paternity uncertainty, tend to exhibit female-biased care.

## Results
### Large variation in sex-difference in parental contributions across different forms of parental care
We found remarkable diversity regarding which sex provides care across the three different care forms in birds (Fig. 1, $N = 1533$ species; Table 1, Supplementary Data 1). At the stage of nest building, both partners built the nest in 938 (61.19%) species (including 56 species of cooperative breeding); the female built the nest alone in 538 species (35.09%); the male built the nest alone in only 57 species (3.72%). At the stage of incubation, female-only care (768 species, 50.10%) and biparental care (740 species, 48.27%) were about equally common, while male-only care was very rare (25 species, 1.63%). At the stage of offspring provisioning, biparental care was the dominant form (1344 species, 87.67%, including 139 species of cooperative breeding), followed by female-only care (166 species, 10.83%), while male-only care (23 species, 1.50%) continued to be the rarest care pattern.

### Moderate consistency of sex-difference in parental care patterns across breeding stages
We revealed moderate consistency of sex-difference in care patterns between temporally consecutive stages of parental care. The direct phenotypical correlations of care patterns between nest building and incubation, between incubation and offspring provisioning, and between nest building and offspring provisioning are all positive (Figs. 2 and 3). Moreover, the multivariate phylogenetic model showed that the three phylogenetic correlations are also positive (nest building −incubation: 0.509 ± 0.051; incubation−offspring provisioning: 0.419 ± 0.050; nest building−offspring provisioning: 0.331 ± 0.055, Fig. 3, Supplementary Data 2: Model 1). To summarize, phenotypic and phylogenetic correlations both suggest that the sex-difference in care patterns across different stages are moderately consistent (Fig. 3). In addition, the multivariate phylogenetic model showed that the sex-difference in care patterns across the three reproductive stages have strong phylogenetic signals ($\lambda = 0.800 ± 0.021$; $0.902 ± 0.016$; $0.769 ± 0.026$, respectively, Fig. 3, Supplementary Data 2: Model 1).

### Biases towards female care under strong sexual selection on males
We quantified the intensity of sexual selection on males by using the principal component 1 (PC1) of the mating system, sexual size dimorphism, and sexual dichromatism as a proxy. Although sexual selection can influence the evolution of all three life history traits, the degree of evolutionary response of each trait may differ greatly. Under certain contexts, sexual selection may cause trait evolution to deviate

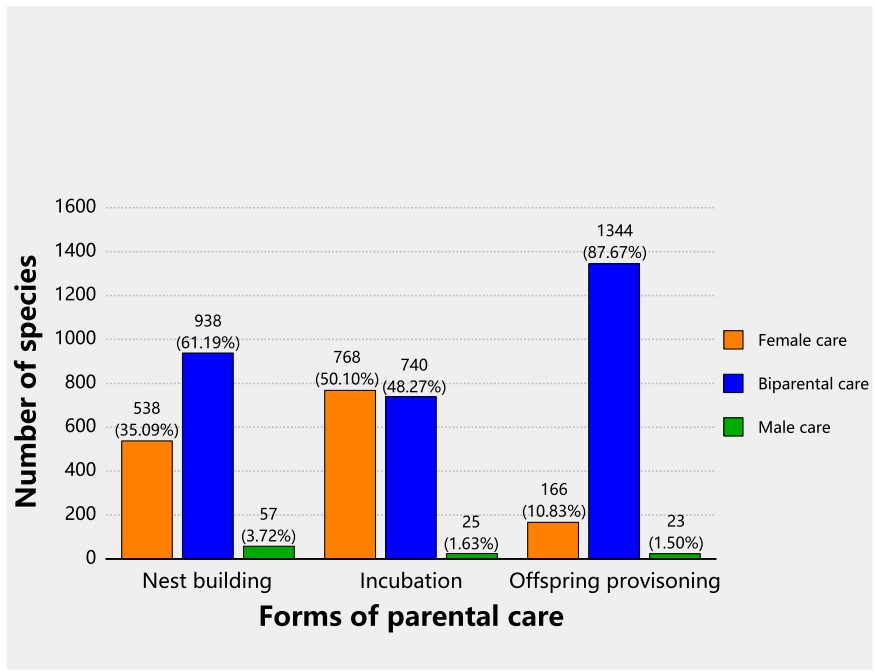

**Fig. 1 | Variation of avian care pattern in different forms across a breeding cycle.** The results were based on 1533 species with full data on care pattern ('Female care', 'Biparental care', and 'Male care') across three forms of parental care: nest building, incubation, and offspring provisioning. Bars of different colors represent different categories of care patterns. Note that a few species of cooperative breeding were grouped into the 'Biparental care' category ($N = 56$ species in nest building, $N = 48$ in incubation, and $N = 139$ in offspring provisioning). Source data for producing this figure can be found in the file "Source Data".

from the expected direction. For example, when males' sexual displays require high agility in the air, sexual selection may produce female-biased sexual size dimorphism because smaller males tend to be more agile[52]. Therefore, instead of using each of the traits alone as a proxy for the intensity of sexual selection, we summarize their evolutionary responses to sexual selection by taking the PC1 of all three traits and use it as a proxy for the intensity of sexual selection. Species with high sexual selection values tend to have males that pair with more than one female and are larger and more colorful than females. In our dataset, the three species with the highest sexual selection scores are the Green peafowl (*Pavo muticus*), the Red junglefowl (*Gallus gallus*), and the Montezuma oropendola (*Psarocolius montezuma*); the three species with the lowest sexual selection scores are the Pheasant-tailed jacana (*Hydrophasianus chirurgus*), the Northern cassowary (*Casuarius unappendiculatus*), and the Common bustard-quail (*Turnix suscitator*).

Overall, our statistical analysis of the phylogenetically controlled regression model (Table 1: Model 1) revealed a significant association between sexual selection and the sex-difference in contribution to nest building, incubation, and offspring provisioning ($t = -7.9$, $p < 0.001$; Fig. 4). The consistent pattern of sexual selection is markedly stronger in the 'Female care' than the 'Biparental care' and 'Male care' category across all three forms of care. The phylogenetic signal was strong in the phylogenetically controlled regression model ($\lambda = 0.59$, 95 CI: 0.47–0.67, Table 1: Model 1). The estimated direction and degree of the fixed effects (i.e., 'form of care', and 'sexual selection') from the linear mixed-effects model were almost the same as the phylogenetically controlled regression model except for the fixed effect 'research effort' which turned to be significant (compare Supplementary Data 1: Model 1 with Table 1: Model 1). In the linear mixed-effects model (Supplementary Data 1: Model 1), sex-difference across three care forms tended to depend on research effort ($t = 2.2$, $p = 0.03$). The random effects of 'family' and 'genus' explained 31% and 9% of the variation in the response variable, respectively, indicating a moderate phylogenetic signal. We further checked the interactions between research effort and sexual selection, and research effort and the three parental care categories ('form of care') directly with additional models. However, none of the interaction terms was significant (Supplementary Data 4).

### Association between the certainty of paternity and male care

Our statistical analyses (Table 1: Model 2 and Model 3) revealed a consistent association between the certainty of paternity and the investment of males in parental care across breeding stages. Species with high levels of extra-pair paternity (EPP) or having broods that contain extra-pair offspring (EPBr) tended to show less male care (i.e., 'Female care' and 'Biparental care' were more abundant; EPP: $t = -3.5$, $p < 0.001$, EPBr: $t = -3.1$, $p = 0.002$; Table 1: Model 2 and Model 3). In the linear mixed-effects model, the estimated direction and degree of extra-pair paternity (either EPP or EPBr) were as strong and significant (EPP: $t = -2.8$, $p = 0.005$, EPBr: $t = -2.3$, $p = 0.02$; Supplementary Data 1: Model 2 and Model 3).

### No clear association between sex-difference in parental care and predation risk

Model 4 did not show a significant association between care pattern and nest daily predation rate across three different care forms ($t = 1.1$, $p = 0.32$ in Table 1: Model 4; $t = 1.3$, $p = 0.18$ in Supplementary Data 1: Model 4). Hence, the difference between males and females in the cost of providing care (predation risk in this case) appeared to be non-essential in determining which sex provides care.

### No clear association between the number of carers and offspring's life history traits

Model 5 showed no significant association between the number of carers in each care form and the nestlings' developmental time ($t = 0.6$, $p = 0.56$ in Table 1: Model 5; $t = 1.4$, $p = 0.16$ in Supplementary Data 1: Model 5). There was also no clear association between the number of carers and the clutch size ($t = -1.3$, $p = 0.18$ in Table 1: Model 5;

**Table 1 | Summary of statistics of five phylogenetically controlled regression models (Model 1 to Model 5) based on the dataset compiled by the original author**

| | | Estimate(β ± SE) | t | p |
|---|---|---|---|---|
| Model 1 (N = 1050) | Random effects: | | | |
| | Phylogeny (λ) | 0.59 (0.47– 0.67) | | |
| | Fixed effects: | | | |
| | Intercept | −0.031 ± 0.157 | −0.2 | – |
| | Nest incubation | −0.178 ± 0.029 | −6.2 | 5.6e−10 |
| | Offspring provisioning | 0.274 ± 0.029 | 9.6 | 0 |
| | Sexual selection | −0.103 ± 0.013 | −7.9 | 2.9e−15 |
| | Research effort | 0.00002 ± 0.00004 | 0.5 | 0.59 |
| Model 2 (N = 180) | Random effects: | | | |
| | Phylogeny (λ) | 0.71 (0.42–0.86) | | |
| | Fixed effects: | | | |
| | Intercept | −0.188 ± 0.188 | −1.0 | – |
| | Nest incubation | −0.225 ± 0.064 | −3.5 | 0.0005 |
| | Offspring provisioning | 0.349 ± 0.070 | 4.9 | 9.6e−07 |
| | EPP | −0.790 ± 0.224 | −3.5 | 0.0005 |
| | Research effort | −0.00003 ± 0.00005 | −0.5 | 0.60 |
| Model 3 (N = 175) | Random effects: | | | |
| | Phylogeny (λ) | 0.69 (0.38–0.86) | | |
| | Fixed effects: | | | |
| | Intercept | −0.183 ± 0.188 | −1.0 | – |
| | Nest incubation | −0.220 ± 0.066 | −3.3 | 0.001 |
| | Offspring provisioning | 0.378 ± 0.074 | 5.1 | 3.4e−07 |
| | EPBr | −0.489 ± 0.157 | −3.1 | 0.002 |
| | Research effort | −0.0000002 ± 0.00006 | 0.004 | 0.97 |
| Model 4 (N = 245) | Random effects: | | | |
| | Phylogeny (λ) | 0.65 (0.40–0.80) | | |
| | Fixed effects: | | | |
| | Intercept | −0.293 ± 0.177 | −1.6 | – |
| | Nest incubation | −0.329 ± 0.053 | −6.2 | 5.6e−10 |
| | Offspring provisioning | 0.459 ± 0.057 | 8.0 | 1.3e−15 |
| | Nest daily predation rate | 1.060 ± 1.059 | 1.0 | 0.32 |
| | Research effort | −0.0001 ± 0.00007 | −1.5 | 0.14 |
| Model 5 (N = 961) | Random effects: | | | |
| | Phylogeny (λ) | 0.68 (0.58–0.75) | | |
| | Fixed effects: | | | |
| | Intercept | 1.521 ± 0.345 | 4.4 | – |
| | Nest incubation | −0.172 ± 0.033 | −5.3 | 1.2e−07 |
| | Offspring provisioning | 0.326 ± 0.033 | 10.1 | 0 |
| | Nestling developmental time | 0.084 ± 0.145 | 0.6 | 0.56 |
| | Clutch size | −0.076 ± 0.057 | −1.3 | 0.18 |
| | Research effort | −0.00004 ± 0.00004 | −0.9 | 0.35 |

For the response variable, Model 1 to Model 4 used the first way of recoding ('Female care', 'Biparental care' and 'Male care' as '−1', '0', and '+1', respectively; species in the 'Cooperative Breeding' category were also coded as '0', because breeders and helpers of both sexes contributed to care). Model 5 used the second way of recoding ('Female care' and 'Male care' as '1', 'Biparental care' as '2', and 'Cooperative Breeding' as '3'). For the random effect (i.e., the phylogenetic tree), the estimated λ is shown. For each fixed effect, the estimate with its standard error (SE), t-value, and corresponding p-value are shown (two-sided tests; no adjustments were made for multiple comparisons). Note that, for each model, we ran the model using 100 different phylogenetic trees from Jetz et al.[94]. Results are therefore based on mean estimates for predictor slopes and model-averaged standard errors.

$t = −1.7$, $p = 0.09$ in Supplementary Data 1: Model 5). These results suggest that the reproductive value of the current brood (represented by clutch size) and brood needs (represented by both clutch size and nestling developmental time) were not determinant factors of the number of individuals that provide care.

## Discussion

Our survey of more than 1500 species of birds revealed that sex-difference in care patterns differ substantially across different care forms (i.e., nest building, incubation, and offspring provisioning). However, statistical testing showed that there is moderate consistency regarding which sex provides care across the three different stages. Using five phylogenetically controlled regression and linear mixed-effects models, respectively, we further tested several ecological and evolutionary factors that may explain sex differences across different forms of parental care, and we identified sexual selection and the certainty of parental care to be the main driving forces. Uniparental care by females tended to be more frequent in species under strong sexual selection on males, and males of species with high certainty of paternity were more likely to contribute to parental care. However, we did not find a significant association between nest predation rate and sex-difference in parental care. There was also no evidence that off-spring's life history traits that reflect their reproductive value and brood needs played a role in the number of carers. Our analyses have been highly robust, as the major findings remain qualitatively unchanged by excluding uncertain species from the dataset (Supplementary Data 3), using a separate dataset compiled by an independent observer (Supplementary Data 5), or using the common entries between two independent datasets (Supplementary Data 6).

### Parental care is not a unitary trait regarding which sex provides care

Our results regarding parental contributions at the stage of offspring provisioning are similar to the findings in Cockburn[53], which did not include data on the stages of nest building and incubation. The marked sex-differences in care patterns across different breeding stages we found in this study (Fig. 1, Table 1 and Supplementary Data 1) indicate that parental care is not a unitary trait regarding which sex provides care but a composite of several integrated features with great variations. Our results and the lack of theoretical predictions on the causes of the variations highlighted important knowledge gaps in our understanding of parental care as a package with several functionally integrated traits, and how males and females were selected to fulfil different roles of parental care in the evolutionary time scale. Studies in birds have identified several factors that affect the (relative) contributions of the male and female parents in the ecological time scale, including the harshness of abiotic environments, especially temperature and rainfall[54–57], predation risk[58,59], the vulnerability of offspring in the absence of parental care[60,61], and the body condition of the parents themselves[62]. Future studies in both empirical and theoretical aspects are needed to investigate the potential for abiotic and biotic factors to impact parental care over evolutionary timescales, the presence of coevolution, as well as the possibility of evolutionary trajectories and evolutionary transitions.

### Moderate consistency of parental care patterns across breeding stages

Despite the remarkable variation in the overall care pattern across different forms of parental care, our statistical analysis showed moderate consistency of sex-difference in parental care between consecutive care forms (Figs. 2, 3 and Supplementary Data 2: Model 1). This result thus adds to the increasing evidence for consistent individual behaviors across social and ecological contexts[25,28,29], and supports the hypothesis that behavioral consistency helps resolving sexual conflict over parental investment[63]. Although specialization leading to

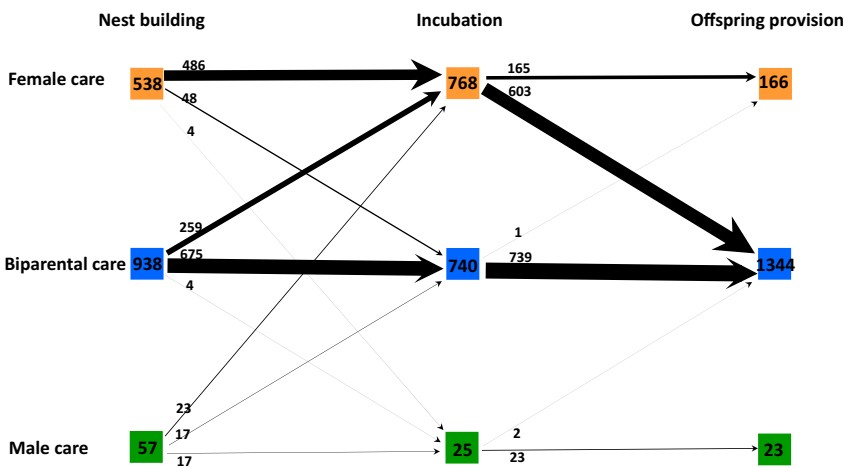

**Fig. 2 | The number of avian species of each parental care pattern (i.e., 'Female care', 'Biparental care', and 'Male care') across different breeding stages from nest building to offspring provisioning.** Source data for producing this figure can be found in the file "Source Data".

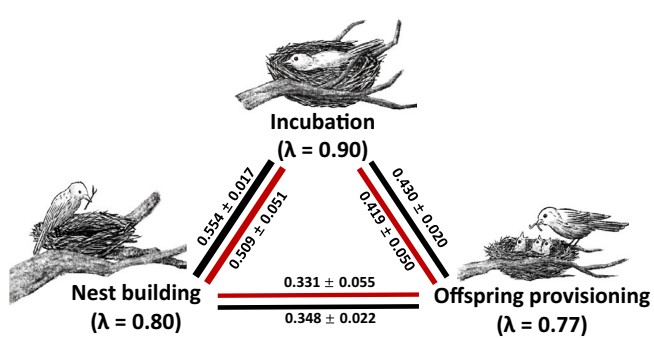

**Fig. 3 | Phenotypical and phylogenetic correlations of care patterns between three distinct parental care stages.** The phenotypical correlations (black) were estimated based on 1533 species. The phylogenetic correlations (red) were estimated using a multivariate phylogenetic model based on 1050 species. Source data for producing this figure can be found in the file "Source Data".

consistency in parental care patterns seems intuitive to expect, the underlying evolutionary mechanisms and whether such specialization provides any consistent fitness advantages are still unclear. Our Model 1 suggests that strong sexual selection on the sex that provides little care may also contribute to the consistent parental care investment of the opposite sex (Table 1). The parental care patterns in the lekking species and sex-role reversed species (Supplementary Table 1) provide the strongest support for this hypothesis. Males of the lekking species such as grouses, paradises and manakins are under extraordinary sexual selection. They have the most extravagant appearances and courtship displays but contribute nothing but genes to the offspring, leaving females to provide all forms of parental care alone[64,65]. But in sex-role reversed species such as jacanas, phalaropes and coucals, where sexual selection on females is stronger than in males, it is the males that are responsible for all forms of parental care duties[66]. Note that the two aforementioned explanations are not mutually exclusive––specialization for caring in one sex and strong sexual selection on the other sex may synergistically cause the former sex to provide care consistently throughout different stages of parental care[67].

### Strong sexual selection was tied to sex-biased care of all forms

Our analyses showed a consistent pattern of sexual selection being stronger in species of female-only care (male-only care in sex-role reversed species) than in species of biparental care across three different forms of parental care. This result is in agreement with the Darwin-Bateman paradigm that predicts sexual selection on males leading to the evolution of conventional care patterns[68], and concurs with the correlation between carotenoid-dependent plumage coloration (generally implies strong sexual selection on males) and reduced male care[69]. It also agrees with a survey of 659 bird species from 113 families, which found that parental cooperation decreased with the intensity of sexual selection and skewed adult sex ratios[70]. The study[70] focused on the association between sexual selection and the "inequality" between males and females in parental care contributions, and therefore they analyzed the parental care data without sex-specificity. Although the parental care data in the previous study contained eight different parental care activities (corresponding to different care forms in our study), the parental cooperation score was calculated by averaging the statistically centered extent of biparental care across the different activities. In contrast, we surveyed more species (1533 species in total) and associated data on sex-difference in parental care in three distinct forms. The two studies are thus complementary to each other, and the combined results suggest that the role of sexual selection on the evolution of sex-biased parental care may be widespread across avian taxa and across different forms of parental care.

### Uncertainty of paternity selected against male care

Our analysis showed a significant association between extra-pair paternity and reduced male care across different parental care forms, in agreement with previous comparative studies with a smaller number of species[71-74]. It contrasts with a recent phylogenetic analysis of the sex roles in birds, which found an increase in male-biased care as the proportion of broods with extra-pair offspring increases, albeit with a relatively small effect size[75]. Although theories generally predict that males should invest more in the care of their genetic offspring and adjust their parental efforts to their share of paternity in the nest[15-17,67], empirical supports have been mixed, with abundant exceptions where males do not seem to react to the loss of paternity by reducing their parental care efforts, such as in dunnocks[76,77], reed buntings[78], and western bluebirds[79]. Recent theoretical studies revealed some conditions where males may evolve to be insensitive to the loss of paternity, e.g., in cooperative breeding species where offspring help to raise their younger (half-)siblings[80], or in the presence of male alternative reproductive tactics where the "sneakier" males specialize in gaining extra-pair paternity[9]. Empirical studies also found that in species where males were not sensitive to paternity loss, paternal care may not be costly in terms of parental survival[78] and/or the loss of opportunities

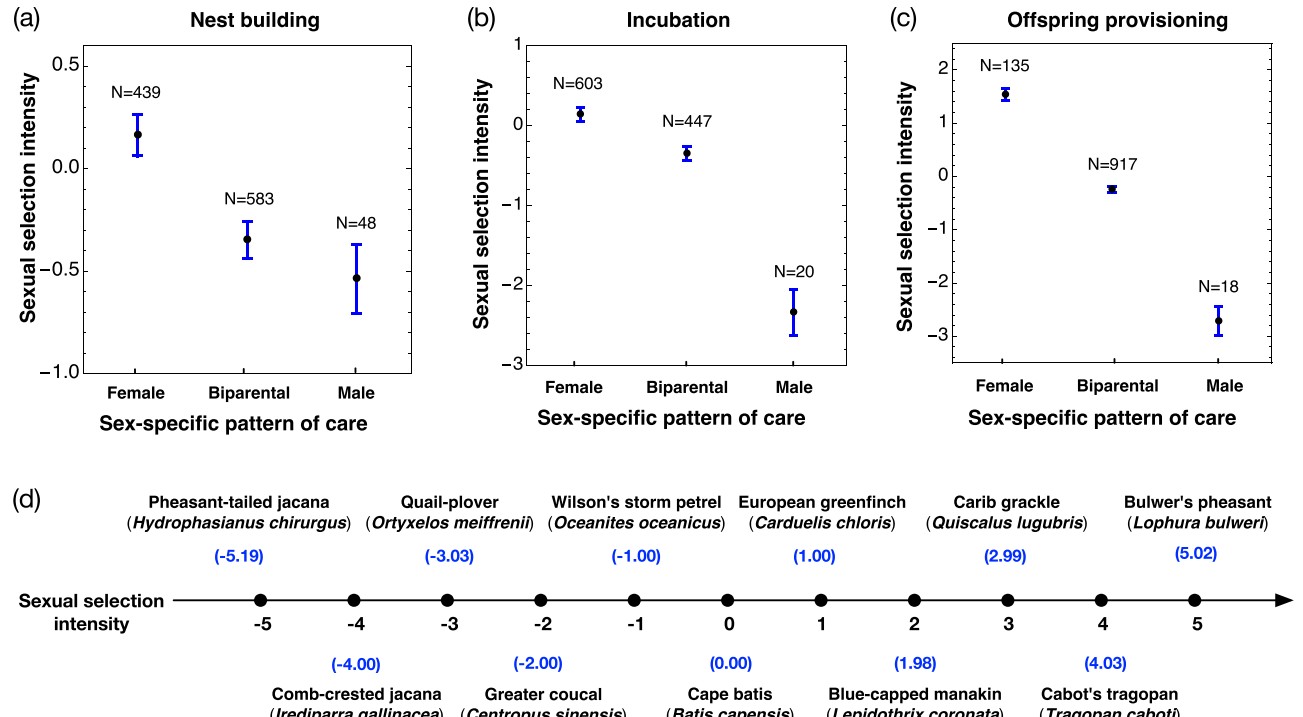

**Fig. 4 | The intensity of sexual selection is strongly associated with care patterns of different forms across breeding stages.** The mean ± standard error of the intensity of sexual selection (measured as the PC1 of the mating system, sexual size dimorphism, and sexual dichromatism) for species of three parental care patterns at the nest building (**a**), incubation (**b**), and offspring provisioning (**c**) stages. **d** A visual presentation of some typical species along the gradient of sexual selection intensity. Species with larger values experience stronger sexual selection on males relative to females. Source data for producing this figure can be found in the file "Source Data".

for siring extra-pair offspring[39]. Few comparative studies (for a rare exception, see Griffin[81]) have tested the role of potential factors that may explain the presence or absence of male response to paternity loss by reducing or withholding paternal care, probably due to a limitation of detailed data on life history traits related to parental care across species. Future efforts in generating and collating such data are therefore indispensable to a better understanding of the relationship between the certainty of paternity and male investment in parental care.

**Nest predation risk did not shape sex-difference in parental care**
Our analyses did not show a significant association between nest predation risk and sex differences in parental care. This result is surprising as it contrasts with a previous survey of 256 species of passerine birds, which found that the frequency of nest visits decreased as the risk of nest predation increased, as frequent bouts of incubation could increase the visibility of a nest[82]. Similar results were found also in seven species of arctic sandpipers[83]. However, it concurs with a recent study which showed that avian species with open nests (considered as providing less protection from predators than closed nests) do not exhibit higher parental cooperation than species with closed nests[84]. Given that the plumage of females is usually drabber and more cryptic than males, we expected species with high nest predation to show more female-biased care. The lack of correlation could be due to either anti-predatory adaptations, confounding factors that masked the effect of female cryptic plumage, or a combination of both. Species that endure high nest predation risk may have evolved strategies that minimize activities that could attract predators, like long on- and off-bouts of incubation[54], and males with brighter plumage may evolve to attend the nest largely at night when visual predators were inactive, such as in the red-capped plover[85]. Confounding factors such as nesting site quality and the shape of nests may also override the advantage of the drabber plumage of females in providing care. For

example, a study using 10 species of open-nesting birds in Arizona, USA revealed a positive correlation between nest predation and parental activity only when nest site effects were considered[86].

**Brood needs and offspring's reproductive value are not associated with the number of care providers**
Since broods of larger sizes and longer nestling developmental time generally have higher needs, we expected that more carers (i.e., both parents relative to a single parent, or breeders and helpers relative to only the breeders) were required to provide the elevated amount of care. But no such association was found in our data. Our results suggested that the amount of parental care a brood receives may not necessarily increase with the number of carers. Indeed, models have shown that a parent may or may not compensate for a reduction of parental effort by the other depending on various factors, including the marginal benefit/harm to offspring as a function of total care received, how well each parent is informed about brood needs, and how well the parents can monitor each other's investment[21,23,24,87]. Negotiation between parents can even produce cases where the offspring do better with one parent than two[22]. Experimental studies by (temporally) removing a parent also showed that the compensation patterns can vary widely from a matching reduction, through no, partial, and full compensation, to even over-compensation[88–91]. Therefore, species are likely to have evolved redundancy in their abilities to provide care, and such abilities could be beneficial to secure reproductive success in cases of losing a partner and/or helper.

**Interesting exceptions to the general patterns of parental care**
Although the main aim of the paper is to depict a general pattern of sex-difference in parental care across breeding stages, we were able to identify several species that do not conform to the general pattern. Among the 1533 avian species we surveyed, a total of 15 species, mainly ritites, tinamous and jacanas, display male-only care throughout the

breeding cycle (Supplementary Table 1). For example, the Wattled jacana (*Jacana jacana*) males lose considerable proportions of paternity when paired with polyandrous females[92], challenging the general rule that the uncertainty of paternity selects against male care. However, this species also features strong sexual selection on females, with females being 48% heavier, more behaviorally dominant, and having greater secondary sexual characters such as fleshy facial ornamentation and wing spurs than males[93]. This conforms to the pattern that strong sexual selection in one sex selects for more care in the opposite sex. The life history features of the exceptional birds provide useful resources for deeper investigations into the driving forces of parental care in the future.

Through a survey of more than 1500 species of birds, we found great diversity in terms of which sex provides care across three different forms (i.e., nest building, incubation, and offspring provisioning). We also found moderate consistency of the sex-difference in care patterns between consecutive stages of care, indicating there may be some shared intrinsic drivers that lead one parent to provide care in different forms. Our models suggest that the intensity of sexual selection is the primary driver of the sex-role variations we found in distinct care forms. We also found that uncertainty of paternity selects against male care. On the whole, our results suggest that parental care should not be treated as a unitary trait, but as a composite of integrated features with great variations. We also identified important knowledge gaps for future theoretical and empirical investigations. For example, we still lack testable theories that make predictions on the relative efforts of male and female parents in different care forms. And we still do not fully understand why males react to a loss of paternity by reducing paternal care in some species but not in others. Would the effects of sexual selection, certainty of paternity, predation risk, and offspring life history traits we found in birds play similar roles in other animal groups? Do other factors, such as adult sex ratio, operational sex ratio, and sex-specific adult mortality, also play a role in shaping sex-role patterns in different forms of parental care? And how do these driving factors interact with each other in eco-evolutionary feedback? Our current work provided a valuable starting point for answering those new questions. We encourage future empirical and theoretical studies to go beyond considering parental care as a unitary trait and delve deeper into its components, such as different forms and stages across a breeding cycle and throughout life.

## Methods
No ethics approval was need in this study.

### Classification of parental care patterns
We surveyed all 1533 bird species in the Birds of the World database[51] for which sex provides parental care in each of the three forms--nest building, incubation, and offspring provisioning--across a reproductive cycle. The three forms (i.e., behaviors of parental care) were chosen due to their representativeness of distinct reproductive stages and the availability of well-documented data across species. We took notes of the parental care features for each species from the breeding section of the species' account and then classified them into four categories for each form of care: (1) 'Male care', where only paternal care was present; (2) 'Female care', where only maternal care was present; (3) 'Biparental care', where both parents provide care, and (4) 'Cooperative Breeding', where helpers of cooperatively breeding species also participate in caring of offspring (typically offspring provisioning). Since we are interested in the overall patterns across different bird species at an evolutionary scale, we did not consider the within-species variations of care patterns in this study. Therefore, cases where a form of care was provided usually by females alone but males were occasionally observed to participate were classified as 'Female care', and vice versa. In rare cases (*N* = 49 species), the parental care information was recorded with uncertain words, such as "reportedly" or

"probably" in one or more care forms (e.g., White-throated Bulbul: nest reportedly built by both sexes; ... incubation possibly by both sexes, period 13 days; chicks fed by both parents). All statistical models were run by first including and then excluding those uncertain data.

In particular, for precocial species in which the young are relatively mature and mobile from hatching (i.e., the young leaves the nest shortly after hatching), although parents usually do not feed the precocial chicks, they still invest intensive care efforts (e.g., leading chicks to the food) until their offspring's independence. In this case, we classified the provisioner sex as the sex who cared for chicks before independence.

In some species of cooperative breeding, the categorization of care patterns in each care form was straightforward (e.g., White helmet-shrike: Cooperative breeder, all group-members assisting in all aspects of nesting duties. Breeding pair chooses nest-site and does most of the construction work, but is assisted by other group members; incubation by all group-members; chicks are brooded and fed by all of the group). In the others, the contribution of helpers to each care form may not be specified. Given that cooperative breeding with helpers usually implies helpers' participation in chick provisioning[53], we classified those species' offspring provisioning as 'Cooperative Breeding', and classified the other two care forms according to additional details in the description. For example, according to the description "Drakensberg rockjumper: breeds as monogamous pair and co-operative, with helpers. Nest built by both sexes, ...; incubation by both sexes; no other information.", we classified this species' nest building and incubation as 'Biparental care', and offspring provisioning as 'Cooperative Breeding'.

Following the above procedures, we collected 1533 species with 'full data' (i.e., information about sex-specific contribution care in all three care forms; 651 non-passerines and 882 passerine species). The scoring of parental care pattern was performed by two observers independently. Both datasets ("species_1533_texts_upload.xlsx", compiled by the original author, and "species_1533_texts_SY_upload.xlsx", compiled by the independent observer) are available in the online repository: https://osf.io/g4xra/. Furthermore, we included the original descriptions of sex-specific parental care contribution (text or their corresponding URLs) for each species in the column 'verbal description of sex-specific parental care 2021' in the datasets. The two independently compiled datasets are highly consistent. The proportion of identical scoring values of sex-specific parental contribution during nest building, incubation, and offspring provision were 92.43% (*N* = 1533 species, *p* < 2.2e−16), 95.82% (*N* = 1533 species, *p* < 2.2e−16), and 93.28% (*N* = 1533 species, *p* < 2.2e−16), respectively. We then matched the scientific names used in the data source[51] (*N* = 1533 species, "species_1533_texts_upload.xlsx" from the online repository: https://osf.io/g4xra/) with the species names from a phylogenetic information source (BirdTree.org)[94]. We included 1410 species where we have complete data on the phylogenetic information and contributor(s) of parental care in nest building, incubation, and offspring provisioning for further analyses using statistical models. To ensure the robustness of our results, we also ran the statistical models using the dataset compiled by the independent observer (Supplementary Data 5) and the common entries between the two independently compiled datasets (Supplementary Data 6).

### Cross-validation with independent datasets
We cross-validated our dataset with two independent data sources (i.e., Cooney et al.[95] and Szekely et al.[96]) and found high consistency between the entries, further ensuring the quality of our data collection.

In the dataset of Cooney et al.[95], there is a single variable that contains information regarding parental care (named 'parental_care_unibi'). The variable has two levels, 'biparental care' and 'uniparental care', respectively. By directly contacting the first author, we learned that the stage of parental care considered in the dataset was

incubation. Therefore, we were able to transform our datasets (i.e., by pooling 'female-only care' and 'male-only care' into 'uniparental care') and compare them with theirs. The results are highly consistent as summarized in Supplementary Table 2.

In the dataset of Szekely et al.[96], there are three variables that we can use to cross-validate our data. We used their 'nest.bld' (relative investment of the sexes in nest building) to compare with our data on nest building; used their 'inc_2' (relative investment of the sexes in incubation of the eggs) to compare with our data on incubation; and used their 'postf.feed_2' (relative investment of the sexes in post-fledging feeding of the offspring) to compare with our data on off-spring provisioning. We converted their entry '0' to 'female-only care', their entry '4' to 'male-only care', and their other entries to 'biparental care' and compared them with the corresponding species in our datasets. The results are also highly consistent as summarized in Supplementary Table 3.

### Explanatory variables in statistical models
(a) We used the PC1 of the mating system, sexual size dimorphism, and sexual dichromatism as a proxy for the intensity of sexual selection ($N = 3025$ species; after accounting for phylogenetic effects and matching with our dataset of sex roles, $N = 1050$ species) to compare sex-difference in parental care across different breeding stages. Spe-cifically, mating systems of passerine species were obtained following Dale et al.[45], and we added non-passerine species that were scored following the same principles. In short, the mating system was scored on a seven-point scale, with '0' representing strict social monogamy (e.g., Zebra finch *Taeniopygia guttata*), '1' representing monogamy with infrequent instances of polygyny observed (<5% of males, e.g., Lazuli bunting *Passerina amoena*) and '−1' representing monogamy with infrequent instances of polyandry observed (<5% of females, e.g., Northern flicker *Colaptes auratus*), '2' representing mostly social monogamy with regular occurrences of facultative social polygyny (5–20% of males, e.g., American redstart *Setophaga ruticilla*) and '−2' representing mostly social monogamy with regular occurrences of facultative social polyandry (5–20% of females, e.g., Pale chanting-goshawk *Melierax canorus*), and '3' representing obligate resource defense polygyny (>20% of males, e.g., Lance-tailed manakin *Chirox-iphia lanceolata*) and '−3' representing obligate resource defense polyandry (>20% of females, e.g., Comb-crested jacana *Irediparra gallinacea*). A small number of species with polygynandrous mating systems were pooled with the monogamous species (e.g., Dunnock, *prunella modularis*). Sexual size dimorphism was estimated by com-bining differences between the sexes in adult body mass, tarsus length, and wing length. In practice, sexual size dimorphism was calculated for three traits representing body size (body mass (g), tarsus length (mm) and wing length (mm)) and was calculated as log (male trait value/female trait value)[97]. Sexual dichromatism was obtained following Gonzalez-Voyer et al.[75]. In short, the mean value of plumage dimorphism is estimated from five body regions (head, back, belly, wings and tail). Plumage was scored using Birds of the World database[51]. The nominate subspecies of each species was scored using plates as the main reference supplemented with images and descrip-tions. A single observer scored each body part separately using the following scheme: −2, the female was substantially brighter and/or more patterned than the male; −1, the female was brighter and/or more patterned than the male; 0, there was no sex difference in the body region or there was a difference but neither could be considered brighter than the other; 1, the male was brighter and/or more pat-terned than the female; 2, the male was substantially brighter and/or more patterned than the female. Thus, positive values represent male-biased ornamentation, zero represents unbiased ornamentation, and negative values represent female-biased ornamentation. The average score of five body regions correlated well with three independent datasets of dichromatism: Spearman rank correlations, rs = 0.705,

$N = 5825$ species[45], rs = 0.867, $N = 905$ species[98], and rs = 0.542, $N = 855$ species[75,99].

(b) EPP was the proportion of extra-pair offspring ($N = 390$ species, after accounting for phylogenetic effects and matching with our dataset of sex roles, $N = 180$ species) and (c) EPBr was the proportion of broods with extra-pair offspring ($N = 386$ species, after accounting for phylogenetic effects and matching with our dataset of sex roles, $N = 175$ species). Data on EPP and EPBr was obtained from the study of Valcu et al. (2021)[100]. (d) Daily nest predation rate (log10 transformed, $N = 580$ species; after accounting for phylogenetic effects and match-ing with our dataset of sex roles, $N = 245$ species) was obtained from Matysioková & Remeš[82], Freeman et al.[101] and Unzeta et al.[102]. (e) Clutch size (log10 transformed, $N = 1270$ species; after accounting for phylogenetic effects and matching with our dataset of sex roles, $N = 961$ species) and (g) length of the nestling developmental period (in days, log10 transformed, $N = 1041$ species; after accounting for phylogenetic effects and matching with our dataset of sex roles, $N = 961$ species) were collated from Cooney et al.[95]. (h) Research effort ($N = 9051$ species; after accounting for phylogenetic effects and matching with our dataset of sex roles, $N = 1376$ species), quantified as the number of independent entries per species in the Zoological Record database[103], was incorporated to account for data quality.

### Statistical analyses
All analyses were carried out within the R statistical environment[104]. Firstly, we summarized the variations of care patterns across nest building, nest incubation and offspring provisioning. This was done by using phylogenetic regression models from the package 'phylolm'[105,106] and linear mixed-effect models from the package 'lme4'[107] with the 'form of care' (three levels: nest building, incuba-tion, and offspring provisioning) as a fixed effect (details see below). Next, we tested whether the sex-specific care patterns are tempo-rally consistent, complementary, or irregular across different breeding stages. To do this, we estimated both direct phenotypical Pearson correlations and phylogenetic correlations between par-ental care patterns of these three reproductive stages. In particular, we implemented a multivariate phylogenetic model from the pack-age 'MCMCglmm'[108] to investigate phylogenetic correlations between care patterns across breeding stages. If the sex-differences in parental care are temporally consistent, we expect positive cor-relations between them. If the sex-differences in parental care are complementary, we expect a negative correlation between the stage of nest building and incubation, followed by a negative correlation between the stage of nest incubation and offspring provisioning. If the sex-differences in parental care are irregular, we expect to observe no clear correlations between stages. Finally, we tried to uncover possible driving forces of the variation of sex differences in parental care across different care forms. This was done by using phylogenetic regression models from the package 'phylolm'[105,106] and linear mixed-effect models from the package 'lme4'[107] with specific fixed effects.

### Two distinct ways to recode the response variables
Considering the differences in premises regarding different hypoth-eses and the number of species available for relevant explanatory variables, we coded the response variables (i.e., the contributor(s) of parental care in each form) in two different ways, depending on the corresponding explanatory variables in a series of models. The first way of recoding the contributor(s) of parental care focuses on which sex provides the care. We recoded 'Female care', 'Biparental care' and 'Male care' as '−1', '0', and '+1', respectively. Species in the 'Cooperative Breeding' category were also coded as '0', because breeders and helpers of both sexes contributed to care. Using this way of recoding, we built four models to test hypotheses regarding the variation/cor-relations of sex-difference in parental care across stages of a breeding

cycle and investigate whether sexual selection, extra-pair paternity, and nest predation were the main driving factors determining which sex provides care in each form. The second way of recoding the contributor(s) of parental care focuses on the number of individuals that provide care to a brood in each of the three forms. In this way, we recoded 'Female care' and 'Male care' as '1', 'Biparental care' as '2', and 'Cooperative Breeding' as '3', since there was one carer (either the male or the female) in the first category, two carers (both the male and female parent) in the second category, and at least three carers (both the male and female breeder and at least one helper) in the third category. The second way of recoding allowed us to build an additional model to test the association between the offspring's life history traits (reflecting the offspring's reproductive value and brood needs) and the number of carers in each care form.

### Linear mixed-effect and phylogenetic models to uncover the variation of care patterns and its possible driving forces

To reveal the possible intrinsic drivers of sex-difference in parental care across different care stages, we implemented five phylogenetic regression models in addition. Detailed information about the five models is listed below.

**Model 1**. The model was built to quantify the association between care pattern in each of the three forms of parental care and sexual selection. In this model, the care pattern (using the first way of recoding) were added as the response variable. We included 'form of care' (three levels: nest building, incubation, and offspring provisioning), 'sexual selection', and 'research effort' as fixed effects. To control for phylogenetic uncertainty, we used a phylogenetically controlled regression method as implemented in the function 'phylolm' from the R package phylolm[105,106]. In this model, we included the phylogenetic tree of species as a random effect and ran the model using 100 different phylogenetic trees[94]. Results are therefore based on mean estimates for predictor slopes and model-averaged standard errors. However, because our dataset included multiple traits for each species (i.e., care pattern in nest building, nest incubation and offspring provisioning) and the three different traits were considered as 'repeated values' in the model, directly adding a phylogenetic tree into the model as a random effect might be unattainable. To solve this issue, we randomly selected one representative observation (out of three) for each species and ran the model with the culled data, which contained only one observation per species. Note that in this way, the full control for phylogeny was achieved at the cost of losing two-thirds of the observations. We additionally used a linear mixed-effects model with the same fixed effects as the phylogenetically controlled regression model. The 'family' and 'genus' of the species were included as a random effect to account for phylogenetic uncertainty. In this way, we have the largest sample size but cannot control for the phylogenetic relationships completely.

**Model 2 and Model 3**. The two phylogenetically controlled regression models were built to assess the association between sex-difference in parental care in each of the three forms of parental care and the degrees of uncertainty in paternity. Care patterns (using the first way of recoding) were added as the response variable. We included 'form of care', 'research effort', and either 'EPP' (in Model 2) or 'EPBr' (in Model 3) as fixed effects. Regarding random effect(s), we treated the phylogenetic tree as a random effect using the 'phylolm'[105,106] package as explained in Model 1. In addition, we built two linear mixed-effects models using the same fixed effects. Regarding random effect(s), due to the limited number of species with data on extra-pair paternity (see the section of 'explanatory variables' above), the model simultaneously contained two random effects ('family' and 'genus') that caused a singular fit issue. This indicated model overfitting, meaning that the random effects structure was too complex to be parameterized by the limited data. Therefore, we only included one random effect (either 'family' or 'genus') that explained more variation of the response variable.

**Model 4**. The model was built to test the association between sex-difference in parental care in each of the three forms of parental care and daily nest predation rates. Care patterns (using the first way of recoding) were added as the response variable. We included 'daily nest predation', 'form of care', and 'research effort' as fixed effects, and the phylogenetic tree as a random effect. In addition, we built a linear mixed-effects model with the same fixed effects while using 'family' and 'genus' of the species as random effects.

**Model 5**. The model was built to test the association between the number of carers in different care forms and the offspring's life history traits. In this model, care patterns (using the second way of recoding) were added as the response variable. We included 'form of care', 'length of nestling developmental period', 'clutch size', and 'research effort' as fixed effects. Like in the other models, the phylogenetic tree was treated as the random effect. In addition, like in Models 1 to 4, we built a linear mixed-effects model where the 'family' and 'genus' of the species were treated as random effects to account for phylogenetic uncertainty.

### Multivariate phylogenetic model estimating phylogenetic correlations

We implemented the multivariate phylogenetic model using the package 'MCMCglmm'[108]. In this model, care patterns of three parental care forms (nest building, nest incubation and offspring provisioning) were added as three response variables. We first randomly selected a tree of species from the BirdTree[94] to include in the multivariate model. This tree of 1050 species was then inversed into a phylogenetic covariance matrix ($n = 1,102,500$ elements) and added as a random effect (similar to including a pedigree matrix as a random effect in an 'Animal Model' in quantitative genetics). For the fixed effects, we included 'sexual selection', and 'research effort'. The three-trait MCMCglmm model was set with a proper prior with all variances set to 0.02, covariances set to zero, and a degree of belief parameter set to v = (size of the matrix +1) = 4. After a burn-in of 10,000 iterations, we ran 260,000 iterations from which a total of 250 samples were drawn (every 1000 iterations). The high thinning interval was required to eliminate temporal autocorrelations between samples. The large phylogenetic covariance matrix and running iterations indicate this model is challenging in terms of the number of parameters to be estimated. By using these multivariate phylogenetic models, we aimed to estimate the 'heritability' (i.e., the 'phylogenetic signal' in this case) of each response variable and the 'genetic correlations' (i.e., the 'phylogenetic correlations' in this case) between the three response variables for further testing whether the parental care patterns are temporally consistent, complementary, or irregular across breeding stages.

### Reporting summary

Further information on research design is available in the Nature Portfolio Reporting Summary linked to this article.

## Data availability

All datasets generated in this study are provided in the Source Data file. Output summaries and summary of statistics from all models can be found in Supplementary Data 1–6. Source data are provided with this paper.

## Code availability

The R code for data analysis and plotting is available in the OSF repository: https://osf.io/g4xra/.

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

## Acknowledgements

We thank Yuansi He and Xinyi Jiang for making the silhouette drawings in Fig. 3. D.W. is funded by the Chinese Academy of Sciences (CAS Pioneer Hundred Talents program), the Third Xinjiang Scientific Expedition Program (Grant No. 2022xjkk0801), and the National Natural Science Foundation of China (32270452). X.-Y.L.R. is funded by the Swiss National Science Foundation (grants 180145 and 211549).

## Author contributions

D.W. and X.-Y.L.R. designed the study; D.W. and S.Y. compiled the datasets; D.W. performed the analysis with help from W.Z. and X.-Y.L.R.; all authors wrote the manuscript and approved the submission of the current version.

## Competing interests

The authors declare no competing interests.
