## [Peer Review File · Nature Communications]

Sex differences in avian parental care patterns vary across the breeding cycleREVIEWER COMMENTS

Reviewer #1 (Remarks to the Author):

In this manuscript, the authors explore the participation of males and females in three typical care forms (nest building, incubation, offspring provisioning) across avian taxa and then test three hypotheses regarding the relationships of sex-specific patterns of care across parental care forms. They additionally examine drivers of sex differences in care forms. The focus on characterizing differences in sex-specific patterns of care across different stages is a much needed and interesting contribution to the field of evolutionary ecology. The findings that 1) sex-specific patterns of care tend to be consistent across care stages and 2) there are links between sex-specific patterns of care and sexual selection (or mating system) is interesting and novel. To the best of my knowledge, no previous work has so thoroughly documented such correlations across species. In general, this is an excellent study. I do have a few comments, though, which I detail below.

Major comments:

Sex role classification: The phrase sex role isn't being used correctly (or as it has been used in previous sexual selection work) in this study. Sex role typically refers to which sex is competitive versus choosy with respect to mates and which sex (if any) provides care. Here, the authors characterize sex roles as male care, female care, biparental care, and cooperation. Sex-specific patterns of care are only one component of sex roles. Thus, throughout the manuscript (i.e. in the Results, Methods, etc.), the 'sex roles' phrasing should be modified and changed to 'sex-specific patterns of care' (or something similar) to be correct. Note: this doesn't impact the conclusions in any way and the results will be equally interesting with this modified phrasing.

Sexual selection metric: Sexual selection in this study is measured as the PC1 of mating system, the sexual size dimorphism, and the sexual dichromatism. I'm a bit unsure of whether this measure is truly a proxy for some component of sexual selection or instead a more general characterization of a mating system. If the latter, I suggest changing the phrasing throughout to reflect that this measure is a characterization of each species' mating system. If the authors do think this measure is a proxy for the strength of sexual selection, additional justification is needed and more details are required to contextualize this proxy. Is sexual selection stronger on male traits versus female traits as this proxy for sexual selection increases? Or something else? In general, if this proxy is a more general characterization of mating systems (rather than sexual selection per se), the conclusions are equally interesting.

Detailed comments:

L45, 'The three main hypotheses focus on the consistency of parental care in temporally consecutive forms': This sentence, as written, is a bit confusing given that the three hypotheses haven't yet been described. Would it be possible to give a brief mention of the three hypotheses above in L41? Or consider rewording this sentence in L45. As is the flow is a bit rough here.

L136-148: It is difficult to know what a high versus low sexual selection value reflects given the metric/proxy of sexual selection used. If the authors choose to continue to refer to this proxy as sexual selection, in this section it will be essential to contextualize what strong versus weak sexual selection refers to. Is this always sexual selection on male traits? Or is this more a characterization of the mating system rather than sexual selection itself (see general comment above)? Again, this is a phrasing thing, but it's important to be precise about this since it influences the general conclusions.

Fig. 4 legend, L15-152: Here, it would be helpful to provide some context of the y-axis scale. Do particular ranges correspond to sexual selection being stronger on male versus female traits? Or something else?

L357: Here, this should say 'In our study, the PCI of.....was used as a proxy for sexual selection strength' given that the PCI of these variables isn't in-of-itself a direct measure of sexual selection on any given trait.

Methods, general: I'll just note that I don't have the expertise to assess how phylogenetic correlations were incorporated into the statistical modeling. That said, the modeling seems very thorough to me.

Reviewer #2 (Remarks to the Author):

This manuscript tests the question of whether sex differences in parental care in birds is consistent across time/modalities. They separated parental care into three categories: nest building, incubation, and provisioning. They found strong correlations between the sex-bias of each form of parental care within species, supporting a "consistent expertise" hypothesis for which sex provides care. They authors next investigated how sexual selection, extra-pair paternity, predation, clutch size and nestling dependency periods impact the sex bias of parental care. This is an interesting paper and impressive effort for collating comparative data, and it moves the field towards a more nuanced view of parental care. However, I am concerned about some of the statistical choices made, and I would like those clarified.

I am concerned about the authors' decision to switch from a fully phylogenetic regression which can be implemented in MCMCglmm, to either a linear model (lme4, insufficient accounting for phylogeny) or a phylogenetic model (phylolm, missing 2/3rds of the observations). MCMCglmm can be used to run multi-variate models where the three forms of care are included as the response variables and their covariances estimated, while fully controlling for phylogeny. This seems to me like it would directly some of the authors' hypotheses about consistency vs switching sex roles. At the least, I think the authors should flip Table 1 and Table S2, so that the phylogenetically controlled models are reported in the main text. The authors should clarify their statistical decisions.

[325-330] My worry is that in species that are 1) not well-studied and 2) less sexually dimorphic, parental care has been more likely to be recorded as female-only even if both sexes contribute (due to biases on the part of historical observers), obscuring male parental effort. Since sexual dimorphism makes up a large part of PC1, I worry this may confound some of the paper's results. The authors do account for research effort based on entries in the zoological record database, including this as a fixed effect in models. However, I think it would be valuable to check the interactions between research effort and sexual selection / the 3 parental care categories directly (especially since in Table 1, Model 1, research effort has a significant effect).

Reviewer #3 (Remarks to the Author):

This paper examines the distribution of three kinds of parental care in birds (nest building, incubation and offspring feeding) and investigates (1) whether the relative contributions of males and females are consistent across the three forms of care; (2) how sex differences in contributions to each form of care are associated with evolutionary and ecological parameters (including the importance of sexual selection, of paternity certainty and of life history parameters).

It shows that (1) there is considerable diversity in sex roles in each component of parental care and that male-only care is much rarer than female-only care or bi-parental care; that bi-parental care is more frequent than female-only care; and that bi-parental care during offspring feeding is more common than in nest building or incubation; (2) there is moderate consistency in sex differences in care across the three forms of parental care.(3) the extent of parental care is associated with

contrasts in many systems and associated variation in the intensity of sexual selection as well as with differences in paternity certainty – but that there is no obvious association with predation risk, the number of carers or the life history traits of offspring. In their Discussion section, the authors emphasised that parental care is not a unitary trait and that the relative contributions of females and males to care vary widely and identify the need for a theoretical framework for exploring the evolution of these differences.

Comparative analysis of sex differences in parental roles in birds is overdue and this represents a useful starting point that will hopefully stimulate both theoretical treatments and further comparative analysis. The results are not particularly surprising – but are certainly reassuring since they indicate that our current understanding of the evolution of sex differences in parental care and their consequences is not too far off the mark though is beyond the scope of this paper, I suspect that it would be useful to explore the exceptions to the generalisations described in this paper and to try to make sense of them – and that this might help to generate new ideas and provide evidence of new relationships.

The analysis is competent and the paper is clear and well organised. There is adequate coverage of the literature. My only serious criticism with the paper is the wording of the 'hypotheses' concerning consistency in the pattern of care used in the Introduction. These include specific causal explanations of the predicted relationships as well as their direction – when the paper only examines the direction of these trends. For example, they refer to the suggestion that there should be consistency in parental contributions by males and females as the 'consistent expertise hypothesis' – but make no effort to investigate whether sex differences in 'expertise' are involved. Similarly, they refer to the idea that the contributions of males and females to care might differ between the stages of reproduction as the 'complementary negotiation hypothesis' – but make no attempt to show that any form of negotiation is involved or to argue that this is likely. Finally, they refer to the idea that sex differences in opportunity costs are responsible for variation in patterns of care as the 'distinct pattern hypothesis' but, once again, do not attempt to show or argue how opportunity costs affect the observed patterns. In each of these cases, there are multiple causal mechanisms that could contribute to the consistency or inconsistency of contributions to parental care across the three stages – and the evidence of consistency that the paper provides no clear evidence of the causal mechanisms involved. It would be helpful to reword the initial predictions in ways that excluded any assumption of the causal mechanisms involved – and to include a more extensive discussion of these in the Discussion.

REVIEWER COMMENTS

Reviewer #1 (Remarks to the Author):

In this manuscript, the authors explore the participation of males and females in three typical care forms (nest building, incubation, offspring provisioning) across avian taxa and then test three hypotheses regarding the relationships of sex-specific patterns of care across parental care forms. They additionally examine drivers of sex differences in care forms. The focus on characterizing differences in sex-specific patterns of care across different stages is a much needed and interesting contribution to the field of evolutionary ecology. The findings that 1) sex-specific patterns of care tend to be consistent across care stages and 2) there are links between sex-specific patterns of care and sexual selection (or mating system) is interesting and novel. To the best of my knowledge, no previous work has so thoroughly documented such correlations across species. In general, this is an excellent study. I do have a few comments, though, which I detail below.

Thanks for your positive comments and helpful suggestions. We have revised the manuscript substantially following the suggestions of you and the other two reviewers. Please see our point-by-point replies to your comments below.

Major comments:

Sex role classification: The phrase sex role isn't being used correctly (or as it has been used in previous sexual selection work) in this study. Sex role typically refers to which sex is competitive versus choosy with respect to mates and which sex (if any) provides care. Here, the authors characterize sex roles as male care, female care, biparental care, and cooperation. Sex-specific patterns of care are only one component of sex roles. Thus, throughout the manuscript (i.e. in the Results, Methods, etc.), the 'sex roles' phrasing should be modified and changed to 'sex-specific patterns of care' (or something similar) to be correct. Note: this doesn't impact the conclusions in any way and the results will be equally interesting with this modified phrasing.

This is a fair point. We agree that the term "sex role" has more components than what we have presented in the previous version of the manuscript. We updated the terminology throughout as suggested. They are highlighted in green colour in the updated manuscript.

Sexual selection metric: Sexual selection in this study is measured as the PC1 of mating system, the sexual size dimorphism, and the sexual dichromatism. I'm a bit unsure of whether this measure is truly a proxy for some component of sexual selection or instead a more general characterization of a mating system. If the latter, I suggest changing the phrasing throughout to reflect that this measure is a characterization of each species' mating system. If the authors do think this measure is a proxy for the strength of sexual

selection, additional justification is needed and more details are required to contextualize this proxy. Is sexual selection stronger on male traits versus female traits as this proxy for sexual selection increases? Or something else? In general, if this proxy is a more general characterization of mating systems (rather than sexual selection per se), the conclusions are equally interesting.

Thanks for the insightful comment. Indeed, the concepts of sexual selection and the mating system have been intertwined since the time of Darwin. At the same time, each of them has evolved to mean sometimes very different things in different contexts (e.g., the meaning of “mating system” in plants and animals, and at the genetic and social levels can be totally different). But we think the two concepts have different emphases in the way that they are generally used in literature — sexual selection is more often considered as an evolutionary force that focuses on the *cause* of a phenomenon, while the mating system is more descriptive and focuses on the *consequence* of sexual selection, although it can certainly feedback to influence the intensity and direction of sexual selection. Because we aim to *identify possible driving forces* that produced the pattern of parental care across breeding stages, we believe that using “sexual selection” here is more appropriate.

As you rightly identified, we quantify the intensity of sexual selection by the PC1 of the mating system, sexual size dimorphism, and sexual dichromatism. We consider it as a proxy for the intensity of sexual selection on males relative to females because although sexual selection can affect each of the three observables, none of them alone is a good proxy for the intensity of sexual selection. For example, we know that high degrees of sexual selection do not necessarily produce a polygynous mating system (Jones & Hunter, 1993; Klug, 2018). Although sex-biased size dimorphism is correlated with the intensity of sexual selection in general (Janicke & Fromonteil, 2021), there are important exceptions in birds, where strong sexual selection actually produces female-biased sexual size dimorphism if the sexual display of males requires high aero-agility (Székely et al., 2004). Similarly, sexual selection does not necessarily produce male-biased sexual dichromatism, due to the genetic correlations between males and females (Dale et al., 2015). Therefore, by taking the PC1 of the three possible consequences of sexual selection (although each of them is not a good proxy), we balance out the effects caused by the exceptions in each of them and generate a more comprehensive picture of the intensity of sexual selection in different species. We thus believe that this quantitative measurement can be used as a suitable proxy for the intensity of sexual selection on males.

We added the following sentences to the revised manuscript (Lines 146–154):

“We quantified the intensity of sexual selection on males by using the principal component 1 (PC1) of the mating system, sexual size dimorphism, and sexual dichromatism as a proxy. Although sexual selection can influence the evolution of all three life history traits, the degree of evolutionary response of each trait may differ greatly. Under special contexts, sexual selection may cause trait evolution to deviate

from the expected direction. For example, when males' sexual displays require high agility in the air, sexual selection may produce female-biased sexual size dimorphism because smaller males tend to be more agile⁴⁸. Therefore, instead of using each of the traits alone as a proxy for the intensity of sexual selection, we summarize their evolutionary responses to sexual selection by taking the PC1 of all three traits and use it as a proxy for the intensity of sexual selection."

References:

- Jones, Ian L., and Fiona M. Hunter. "Mutual sexual selection in a monogamous seabird." *Nature* 362.6417 (1993): 238-239.
- Klug, Hope. "Why monogamy? A review of potential ultimate drivers." *Frontiers in Ecology and Evolution* 6 (2018): 30.
- Janicke, Tim, and Salomé Fromontel. "Sexual selection and sexual size dimorphism in animals." *Biology Letters* 17.9 (2021): 20210251.
- Székely, Tamás, Robert P. Freckleton, and John D. Reynolds. "Sexual selection explains Rensch's rule of size dimorphism in shorebirds." *Proceedings of the National Academy of Sciences* 101.33 (2004): 12224-12227.
- Dale, James, et al. "The effects of life history and sexual selection on male and female plumage colouration." *Nature* 527.7578 (2015): 367-370.

Detailed comments:

L45, 'The three main hypotheses focus on the consistency of parental care in temporally consecutive forms': This sentence, as written, is a bit confusing given that the three hypotheses haven't yet been described. Would it be possible to give a brief mention of the three hypotheses above in L41? Or consider rewording this sentence in L45. As is the flow is a bit rough here.

Indeed. Considering your suggestion and that of Reviewer 3, we changed the sentence to the following (Lines 41–44):

"We then explore the temporal consistency of sex-specific contribution between consecutive stages of parental care. In other words, if a male has built the nest, would he continue to incubate the eggs, or leave the task to the female? Similarly, if a female has incubated the eggs, would she continue to feed the chicks after they hatch, or leave them to the male?"

We also thoroughly rewrote the following paragraph (Lines 46–66) to explain the three hypotheses more clearly.

L136-148: It is difficult to know what a high versus low sexual selection value reflects given the metric/proxy of sexual selection used. If the authors choose to continue to refer to this proxy as sexual selection, in this section it will be essential to contextualize what strong versus weak sexual selection refers to. Is this always sexual selection on male traits? Or is this more a characterization of the mating system rather than sexual selection itself (see general comment above)? Again, this is a phrasing thing, but it's

important to be precise about this since it influences the general conclusions.

Good point. We added the following sentence to explain what a high/low sexual selection value means (Lines 154–155):

“Species with high sexual selection values tend to have males that pair with more than one female and are larger and more colourful than females.”

Fig. 4 legend, L15-152: Here, it would be helpful to provide some context of the y-axis scale. Do particular ranges correspond to sexual selection being stronger on male versus female traits? Or something else?

Excellent suggestion. We now include a new panel (d) in the revised Figure 4 to provide visual context for the intensity of sexual selection on males relative to females across species. We also added the following sentences to the manuscript (Lines 155–159):

“In our dataset, the three species with the highest sexual selection scores are the green peafowl (*Pavo muticus*), the red junglefowl (*Gallus gallus*), and the Montezuma oropendola (*Psarocolius montezuma*); the three species with the lowest sexual selection scores are the pheasant-tailed jacana (*Hydrophasianus chirurgus*), the northern cassowary (*Casuaris unappendiculatus*), and common bustard-quail (*Turnix suscitator*).”

Figure 4. The mean ± standard error of the intensity of sexual selection (measured as the PC1 of the mating system, sexual size dimorphism, and sexual dichromatism) for species of three parental care patterns at the nest building (a), incubation (b), and offspring provisioning (c) stages. (d) A visual presentation of some typical species along the gradient of sexual selection intensity. Species with larger values experience stronger

sexual selection on males relative to females. Photo credits: Pheasant-tailed jacana: Charles J Sharp; Comb-crested jacana: Julie Burgher; Quail-plover: Nigel Voaden; Greater coucal: David V. Raju; Wilson's storm petrel: John J. Harrison; Cape batis: Oom Kosie; Greenfinch: Charles J. Sharp; Blue-capped manakin: Ltoniolo (Wikimedia username); Carib grackle: Tom Friedel; Cabot's tragopan: Hectonichus (Wikimedia username); Bulwer's pheasant: Stickpen (Wikimedia username). All photos are in the public domain.

L357: Here, this should say 'In our study, the PC1 of.....was used as a proxy for sexual selection strength' given that the PC1 of these variables isn't in-of-itself a direct measure of sexual selection on any given trait.

We adopted your suggestion (Lines 405–406).

Methods, general: I'll just note that I don't have the expertise to assess how phylogenetic correlations were incorporated into the statistical modeling. That said, the modeling seems very thorough to me.

We thank the reviewer again for the constructive critics.

Reviewer #2 (Remarks to the Author):

This manuscript tests the question of whether sex differences in parental care in birds is consistent across time/modalities. They separated parental care into three categories: nest building, incubation, and provisioning. They found strong correlations between the sex-bias of each form of parental care within species, supporting a "consistent expertise" hypothesis for which sex provides care. They authors next investigated how sexual selection, extra-pair paternity, predation, clutch size and nestling dependency periods impact the sex bias of parental care. This is an interesting paper and impressive effort for collating comparative data, and it moves the field towards a more nuanced view of parental care. However, I am concerned about some of the statistical choices made, and I would like those clarified.

Thanks for your positive assessments and constructive suggestions to improve our paper. To address the concerns regarding the statistical choices, we followed your suggestions and added some additional statistical analyses and clarified the choices we made. Please see our point-by-point reply below.

I am concerned about the authors' decision to switch from a fully phylogenetic regression which can be implemented in MCMCglmm, to either a linear model (lme4, insufficient accounting for phylogeny) or a phylogenetic model (phylolm, missing 2/3rds of the

observations). MCMCglmm can be used to run multi-variate models where the three forms of care are included as the response variables and their covariances estimated, while fully controlling for phylogeny. This seems to me like it would directly some of the authors' hypotheses about consistency vs switching sex roles. At the least, I think the authors should flip Table 1 and Table S2, so that the phylogenetically controlled models are reported in the main text. The authors should clarify their statistical decisions.

Thanks for your comments. We would like first to clarify a possible misunderstanding. In fact, we implemented MCMCglmm to test whether the sex-specific patterns of care are temporally consistent across different stages, precisely as you suggested. The linear and phylogenetic models were instead used for different purposes (e.g., to reveal the phenotypic variation of sex-specific care patterns across different breeding stages). Each of the modelling approaches has some advantages and disadvantages. The linear model is the most robust one with the largest sample size but cannot control for the phylogenetic relationships completely. The phylogenetic model can control for the phylogenetic uncertainty in a better way but loses two-thirds of the observations. However, both models can reveal the phenotypical variation in the sex-specific patterns of care across breeding stages. The MCMCglmm model can control for the phylogenetic uncertainty as well (note that only one phylogenetic tree was included). However, this method does not allow us to compare the differences regarding which sex provides care across three reproductive stages. Nevertheless, the three different types of models all consistently showed that strong sexual selection was tied to sex-biased care of all forms (please see Table 1, Table S1 and Table S2).

To prevent further confusion, we reorganized the *Methods* section to mention the MCMCglmm approach in the first paragraph under the subsection "*Statistical analyses*" (Lines 454–460):

"In particular, we implemented a multivariate phylogenetic model from the package 'MCMCglmm'¹⁰⁰ to investigate phylogenetic correlations between sex-specific care patterns. If the sex-specific care patterns are temporally consistent, we expect positive correlations between them. If the sex-specific care patterns are complementary, we expect a negative correlation between the stage of nest building and incubation, followed by a negative correlation between the stage of nest incubation and offspring provisioning. If the sex-specific care patterns are irregular, we expect to observe no clear correlations between stages."

More details about this method were provided in the subsection "*Multivariate phylogenetic model estimating phylogenetic correlations*" (Lines 538–555).

We also adopted your suggestion to flip Table 1 and Table S2 so that the phylogenetically controlled models are reported in the main text. The corresponding statistical results in the *Results* section are updated, too. Please see the revised manuscript and the supplementary tables for details.

[325-330] My worry is that in species that are 1) not well-studied and 2) less sexually dimorphic, parental care has been more likely to be recorded as female-only even if both sexes contribute (due to biases on the part of historical observers), obscuring male parental effort. Since sexual dimorphism makes up a large part of PC1, I worry this may confound some of the paper's results. The authors do account for research effort based on entries in the zoological record database, including this as a fixed effect in models. However, I think it would be valuable to check the interactions between research effort and sexual selection / the 3 parental care categories directly (especially since in Table 1, Model 1, research effort has a significant effect).

Thanks for your comments. As suggested, we ran several additional models to check the interactions between research effort and sexual selection/the three parental care categories directly (Model 1), we also checked the interactions in the other four models (Model 2 to Model 5), but none of those interaction terms was statistically significant. Detailed statistical results are now included in supplementary Table S4.

We thank the reviewer again for the detailed reviews, and especially for helping us strengthen the statistical analysis of this work.

Reviewer #3 (Remarks to the Author):

This paper examines the distribution of three kinds of parental care in birds (nest building, incubation and offspring feeding) and investigates (1) whether the relative contributions of males and females are consistent across the three forms of care; (2) how sex differences in contributions to each form of care are associated with evolutionary and ecological parameters (including the importance of sexual selection, of paternity certainty and of life history parameters).

It shows that (1) there is considerable diversity in sex roles in each component of parental care and that male-only care is much rarer than female-only care or bi-parental care; that bi-parental care is more frequent than female-only care; and that bi-parental care during offspring feeding is more common than in nest building or incubation; (2) there is moderate consistency in sex differences in care across the three forms of parental care. (3) the extent of parental care is associated with contrasts in many systems and associated variation in the intensity of sexual selection as well as with differences in paternity certainty – but that there is no obvious association with predation risk, the number of carers or the life history traits of offspring. In their Discussion section, the authors emphasised that parental care is not a unitary trait and that the relative contributions of females and males to care vary widely and identify the need for a theoretical framework for exploring the evolution of these differences.

Comparative analysis of sex differences in parental roles in birds is overdue and this

represents a useful starting point that will hopefully stimulate both theoretical treatments and further comparative analysis. The results are not particularly surprising – but are certainly reassuring since they indicate that our current understanding of the evolution of sex differences in parental care and their consequences is not too far off the mark though is beyond the scope of this paper, I suspect that it would be useful to explore the exceptions to the generalisations described in this paper and to try to make sense of them – and that this might help to generate new ideas and provide evidence of new relationships.

Thanks for your encouraging words. We fully agree that the exceptions identified in this work can become valuable candidates for deeper investigations into the underlying mechanisms of sex-specific patterns of parental care in the future. To highlight this point and encourage future research, we added a list of 15 species (Supplementary Table S5) that are featured with male-only care, and included the following subsection (Lines 329–340) to the *Discussion*:

“Interesting exceptions to the general patterns of sex-specific care

Although the main aim of the paper is to depict a general pattern of sex-specific parental care across breeding stages, we were able to identify several species that do not conform to the general pattern. In particular, among the 1533 avian species we surveyed, 15 species, mainly rittites, tinamous and jacanas, display male-only care throughout the breeding cycle (Table S5). For example, the wattled jacana (*Jacana jacana*) males lose considerable proportions of paternity when paired with polyandrous females⁸⁶, challenging the general rule that the uncertainty of paternity selects against male care. However, the species also features strong sexual selection on females, with females being 48% heavier, more behaviorally dominant, and having greater secondary sexual characters such as fleshy facial ornamentation and wing spurs than males⁸⁷, thus conforming to the pattern that strong sexual selection in one sex selects for more care in the opposite sex. The life history features of the exceptional birds provide useful resources for deeper investigations into the driving forces of sex-specific patterns of parental care in the future.”

The analysis is competent and the paper is clear and well organised. There is adequate coverage of the literature. My only serious criticism with the paper is the wording of the ‘hypotheses’ concerning consistency in the pattern of care used in the Introduction. These include specific causal explanations of the predicted relationships as well as their direction – when the paper only examines the direction of these trends. For example, they refer to the suggestion that there should be consistency in parental contributions by males and females as the ‘consistent expertise hypothesis’ – but make no effort to investigate whether sex differences in ‘expertise’ are involved. Similarly, they refer to the idea that the contributions of males and females to care might differ between the stages of reproduction as the ‘complementary negotiation hypothesis’ – but make no attempt to show that any form of negotiation is involved or to argue that this is likely. Finally, they refer to the idea that sex differences in opportunity costs are responsible for variation in

patterns of care as the 'distinct pattern hypothesis' but, once again, do not attempt to show or argue how opportunity costs affect the observed patterns. In each of these cases, there are multiple causal mechanisms that could contribute to the consistency or inconsistency of contributions to parental care across the three stages – and the evidence of consistency that the paper provides no clear evidence of the causal mechanisms involved. It would be helpful to reword the initial predictions in ways that excluded any assumption of the causal mechanisms involved – and to include a more extensive discussion of these in the Discussion.

Thanks for your positive evaluation and constructive suggestions. It is true that our statistical models can only establish correlations but not prove causal relationships. We took your criticism seriously and followed your suggestion to revise the manuscript throughout.

Specifically, we replaced the sentence in the 2nd paragraph “We then test three main hypotheses regarding the relationships of sex roles across distinct parental care forms” with the following (Lines 41–44):

“We then explore the temporal consistency of sex-specific contribution between consecutive stages of parental care. In other words, if a male has built the nest, would he continue to incubate the eggs, or leave the task to the female? Similarly, if a female has incubated the eggs, would she continue to feed the chicks after they hatch, or leave them to the male?”

We also rewrote the 3rd paragraph (Lines 46–66) thoroughly and made it clear that the results of our statistical models are correlational, and do not support or disprove any specific causal pathway. For example:

“Here, we **statistically test** whether the temporal consistency of parental care follows one of the **three patterns**, namely, being consistent (**positive correlation** between stages), complementary (**negative correlation** between stages), or irregular (**no significant correlation** between stages) ... (Lines 46–48) ... Given the tantalizing evidence pointing toward all three possibilities, we build **statistical models** to find the general temporal consistency pattern of parental care across avian taxa, which may aid **future studies that aim to uncover the causal pathways** leading to the stage-specific parental care pattern in different bird species. (Lines 63–66)

Finally, we revised the 3rd paragraph of the *Discussion* to discuss two possible causal mechanisms (i.e., strong sexual selection on one sex and specialization in caring by the opposite sex) that may synergistically contribute to the consistency of sex-specific parental contribution across different breeding stages.

We thank the reviewer again for the constructive critics and for sharing his/her insightful thoughts.

REVIEWER COMMENTS

Reviewer #1 (Remarks to the Author):

This authors have thoroughly revised their manuscript in response to earlier comments from me and other referees. All of my earlier comments have been sufficiently addressed, and this work will make a strong and important contribution to the field.

Reviewer #4 (Remarks to the Author):

I was not one of the previous Reviewers, thus I had a fresh looked at the MS and the Responses. I entirely agree with the Authors that avian care is a great system to understand care evolution in general, and beyond as a major component of sex roles.

The MS puts together an impressive amount of data (see below), the analyses are thorough and the conclusions are fairly solid. I've also checked the Responses and I'm fine with the Authors responses and with their follow-up actions.

Nevertheless, the revised MS falls short of my expectations on 2 accounts:

1. The data quality is patchy. Since the care response variables were generated by scoring verbal descriptions from literature, the readers want an assurance that these scores are reliable and reflect the parental care of a given species. Repeatability, eg scoring the same text by independent observers, is one potential solution.

The other solution is comparing the authors' dataset with publicly available comprehensive datasets and report the consistency with these independent datasets (eg Long et al. 2022 Behav Ecol Soci Biol 76:92, Cooney et al. 2022 Nat Comms, Szekely et al. 2022 Dryad Dataset <https://doi.org/10.5061/dryad.fbg79cnw7>)

I also noted that the Authors used only a subset of available data. For instance, extra-pair paternity data are available for over 280 bird species (Valcu et al 2021 Mol Ecol) whereas the MS seems to use only 160 species. Similarly, nest predation is a widely reported measure whereas only a few 100 species were analysed (largely from passerines) in the MS; I guess, data must be available for at least 800 bird species.

2. The strength of the paper is exploring the associations among care components and linking these to other components of sex roles, eg mating systems (see comments by Rev 1). Whilst the novelty of the work is in the large dataset and the clear presentation of the work, previous studies that explored similar associations should be credited (eg Long et al. 2022, Gonzalez-Voyer et al. 2022).

In my view, the ecological and life-history predictors used in the MS are quite simplistic and given the huge complexity of both types of variables (ie ecology and life history), using a limited set of proxies for these can only uncover preliminary associations.

Dear Editor and reviewers, we have now thoroughly revised the manuscript following the comments and suggestions of Reviewer 4. Our responses are in green text in the point-by-point response letter.

Reviewer #1 (Remarks to the Author):

This authors have thoroughly revised their manuscript in response to earlier comments from me and other referees. All of my earlier comments have been sufficiently addressed, and this work will make a strong and important contribution to the field.

We thank the reviewer for the encouraging assessment and constructive comments that helped improve our manuscript.

Reviewer #4 (Remarks to the Author):

I was not one of the previous Reviewers, thus I had a fresh looked at the MS and the Responses. I entirely agree with the Authors that avian care is a great system to understand care evolution in general, and beyond as a major component of sex roles.

The MS puts together an impressive amount of data (see below), the analyses are thorough and the conclusions are fairly solid. I've also checked the Responses and I'm fine with the Authors responses and with their follow-up actions.

Thank you for acknowledging the merits of our manuscript.

Nevertheless, the revised MS falls short of my expectations on 2 accounts:

1. The data quality is patchy. Since the care response variables were generated by scoring verbal descriptions from literature, the readers want an assurance that these scores are reliable and reflect the parental care of a given species. Repeatability, eg scoring the same text by independent observers, is one potential solution.

Thanks for the suggestion. We followed your advice and asked an additional observer (now included as a coauthor in the revised manuscript) to perform the scoring independently. After that, we compared the two datasets by calculating the proportion of the same scoring values. The results are highly consistent as summarized in the following table. In addition, we ran the statistical models using the dataset compiled by the independent observer (Table S6) and the common entries between the two independently compiled datasets (Table S7). All our major findings remained qualitatively unchanged in the additional analyses. We have now added this information to the revised manuscript (tracked version, Lines 223-226, Lines 406-414).

Comparison between the datasets compiled by the original author and the independent observer			
Form of parental care	Repeatability (proportion of the same scoring values)	Sample size	p-value
Nest building	0.9243	1533	<0.00001
Incubation	0.9582	1533	<0.00001
Offspring provisioning	0.9328	1533	<0.00001

Both datasets (“species_1533_texts_upload.xlsx”, compiled by the original author, and “species_1533_texts_SY_upload.xlsx”, compiled by the independent author) are available in the online repository: <https://osf.io/g4xra/>. Furthermore, we included the original sex role descriptions (text or their corresponding URLs) for each species in the column ‘verbal description of sex-specific parental care 2021’ in the datasets. Readers can conveniently double-check our scoring of the verbal descriptions by following the same scoring procedures and rules, and reanalyze the data when needed.

The other solution is comparing the authors' dataset with publicly available comprehensive datasets and report the consistency with these independent datasets (eg Long et al. 2022 Behav Ecol Soci Biol 76:92, Cooney et al. 2022 Nat Comms, Szekely et al. 2022 Dryad Dataset <https://doi.org/10.5061/dryad.fbg79cnw7>).

Another good idea. We carefully checked the data sources suggested by the reviewer and managed to compare our datasets with two independent sources. Please see the details below.

In the dataset of Cooney et al. (2020), there is a single variable that contains information regarding parental care (named 'parental_care_unibi'). The variable has two levels, 'biparental care' and 'uniparental care', respectively. By directly contacting the first author, we learned that the stage of parental care considered in the dataset was incubation. Therefore, we were able to transform our datasets (i.e., by pooling 'female-only care' and 'male-only care' into 'uniparental care') and compare them with theirs. The results are highly consistent as summarized in the following table.

Incubation, compared with Cooney et al. (2020)			
Our dataset	Pearson correlation	Sample size	p-value
species_1533_texts_upload.xlsx	0.8144	991	<0.00001
species_1533_texts_SY_upload.xlsx	0.8138	991	<0.00001

In the dataset of Long et al. (2022), the investments of each sex into nest building, nest guarding, and incubation were pooled into a variable called "pre-hatching care", and the investments of each sex into chick brooding, feeding, guarding, post-fledging feeding and guarding were pooled together into a variable called "post-hatching care". Due to the merging, rescaling, and transformation of the original data in their dataset, we were not able to compare it with ours. However, we noted that the dataset of Long et al. (2022) and the dataset of Szekely et al. (2022) shared the same original scoring values of parental care. Thus, we next compared our datasets with Szekely et al. (2022).

In the dataset of Szekely et al. (2022), there are three variables that we can use to cross-validate our data. We used their 'nest.bld' (relative investment of the sexes in nest building) to compare with our data on nest building; used their 'inc_2' (relative investment of the sexes in incubation of the eggs) to compare with our data on incubation; and used their 'postf.feed_2' (relative investment of the sexes in post-fledging feeding of the offspring) to compare with our data on offspring provisioning. We converted their entry '0' to 'female-only care', their entry '4' to 'male-only care', and their other entries to 'biparental care' and compared them with the corresponding species in our datasets. The results are also highly consistent as summarized in the following table.

Nest building, compared with Szekely et al. (2022)			
Our dataset	Pearson correlation	Sample size	p-value
species_1533_texts_upload.xlsx	0.8564	1035	<0.00001
species_1533_texts_SY_upload.xlsx	0.8558	1035	<0.00001
Incubation, compared with Szekely et al. (2022)			
Our dataset	Pearson correlation	Sample size	p-value
species_1533_texts_upload.xlsx	0.9229	1083	<0.00001
species_1533_texts_SY_upload.xlsx	0.9359	1083	<0.00001
Offspring provisioning, compared with Szekely et al. (2022)			
Our dataset	Pearson correlation	Sample size	p-value
species_1533_texts_upload.xlsx	0.7297	408	<0.00001
species_1533_texts_SY_upload.xlsx	0.6134	408	<0.00001

Despite the high consistency between the datasets, the agreements between datasets during the offspring provisioning stage are slightly lower than during the other two stages. This is because we included the cases where parents of precocial species leading their chicks to food before they reach independence, which was not considered as post-fledging feeding in the dataset of Székely et al. (2022).

To summarize, we were able to cross-validate our dataset with high consistency between the entries with two independent data sources, further ensuring the quality of our data collection.

References

- Long, Xiaoyan, et al. "Does ecology and life history predict parental cooperation in birds? A comparative analysis." *Behavioral Ecology and Sociobiology* 76.7 (2022): 92.
- Cooney, Christopher R., et al. "Ecology and allometry predict the evolution of avian developmental durations." *Nature Communications* 11.1 (2020): 2383.
- Székely, Tamas et al. (2022), Sex roles in birds: influence of climate, life histories and social environment, Dryad, Dataset, <https://doi.org/10.5061/dryad.fbg79cnw7>

I also noted that the Authors used only a subset of available data. For instance, extra-pair paternity data are available for over 280 bird species (Valcu et al 2021 Mol Ecol) whereas the MS seems to use only 160 species.

Thank you for the comment. We used a subset of available data for two reasons.

First, the Valcu et al (2021) paper was not available when we submitted the manuscript, and therefore, we only used the dataset of Brouwer & Griffith (2019), which was the most complete dataset back then, during the original analysis. Now we have incorporated the additional data from Valcu et al. (2021) and updated the results of our analysis. Thank you for this very useful suggestion.

Second, matching our dataset of sex roles with phylogenetic trees to account for phylogenetic effects also reduced the number of species that can be used for the analysis (i.e., phylogenetic models). Specifically, the dataset of Brouwer & Griffith (2019) contains 329 species with information on the proportion of extra-pair offspring (EPP) and 334 species with information on the proportion of broods with extra-pair offspring (EPBr). After accounting for the phylogenetic effects using the data source of BirdTree.org (Jetz et al., 2012) and matching the species entries with our dataset of sex roles, we only had 160 species of EPP and 162 species of EPBr for the analysis of Model 2 and Model 3, respectively.

Following the reviewer's suggestion, we updated Model 2 and Model 3 using the more complete (and to our knowledge, the latest) dataset from Valcu et al. (2021). In this dataset, there are 390 species with information on the proportion of EPP and 386 species with information on the proportion of EPBr. After accounting for phylogenetic effects and matching with our dataset of sex roles, we had 180 species with EPP information and 175 species with EPBr information, respectively. The output of the phylogenetically controlled Model 2 and Model 3 in the updated manuscript is listed below.

		Estimate ($\beta \pm SE$)	t	p
Model 2	Random effects:			
	Phylogeny (λ)	0.71 (0.42 – 0.86)		
	Fixed effects:			
	Intercept	-0.188 \pm 0.188	-1.0	-
	Nest incubation	-0.225 \pm 0.064	-3.5	<0.001
	Offspring provisioning	0.349 \pm 0.070	4.9	<0.001

	EPP	-0.790 ± 0.224	-3.5	<0.001
	Research effort	-0.00003 ± 0.00005	-0.5	0.6
Model 3	Random effects:			
	Phylogeny (λ)	0.69 (0.38 – 0.86)		
	Fixed effects:			
	Intercept	-0.183 ± 0.188	-1.0	-
	Nest incubation	-0.220 ± 0.066	-3.3	0.01
	Offspring provisioning	0.378 ± 0.074	5.1	<0.001
	EPBr	-0.489 ± 0.157	-3.1	0.002
	Research effort	-0.0000002 ± 0.00006	0.004	0.97

Same as for the old models, we found a consistent association between the certainty of paternity and male parental care across breeding stages. Species with high levels of EPP or EPBr tended to show less male care. However, the effect size of EPBr turned out to be significant in the updated model ($t=-3.1$, $p=0.002$), while it was not significant in the old model ($t=-1.5$, $p=0.15$). The new analyses confirmed and strengthened the robustness of our results. We updated the corresponding results in the text (Lines 193-196; Lines 453-456), Table 1 in the revised manuscript, and Table S1 in the supplementary information.

References:

- Valcu, Cristina-Maria, Mihai Valcu, and Bart Kempenaers. "The macroecology of extra-pair paternity in birds." *Molecular Ecology* 30.19 (2021): 4884-4898.
- Brouwer, Lyanne, and Simon C. Griffith. "Extra-pair paternity in birds." *Molecular Ecology* 28.22 (2019): 4864-4882.
- Jetz, Walter, et al. "The global diversity of birds in space and time." *Nature* 491.7424 (2012): 444-448.

Similarly, nest predation is a widely reported measure whereas only a few 100 species were analysed (largely from passerines) in the MS; I guess, data must be available for at least 800 bird species.

Indeed, similar to the previous concern you raised, the number of species with information on nest predation rate (used in our Model 4) was mainly limited by matching our dataset with phylogenetic trees accounting for phylogenetic effects. We initially compiled daily nest predation rate data for 486 avian species, obtained from Matysioková and Remeš (2018) and Unzeta et al. (2020). After accounting for phylogenetic effects and matching with our dataset of sex roles (1410 species), we were left with 225 species that can be used for Model 4.

To expand the sample size as the reviewer suggested, we carefully checked the literature again and managed to add more species including some non-Passeriformes to the dataset. Now, our original dataset on daily nest predation rates contains 580 species, compiled from Matysioková and Remeš (2018), Unzeta et al. (2020), and Freeman et al. (2020). After accounting for phylogenetic effects and matching with our dataset on sex roles, we now have 245 species for running Model 4 (see below). Relevant results are now updated in the text (Line 200; Lines 455-456) and Table 1 in the revised manuscript, as well as Table S1 in the supplementary materials. The new output of the updated model showed the same pattern as we found before, namely, there was no significant association between nest predation risk and sex-specific parental care investment in birds.

Estimate ($\beta \pm SE$)	t	p
----------	----------

Model 4	Random effects:			
	Phylogeny (λ)	0.65 (0.40 – 0.80)		
	Fixed effects:			
	Intercept	-0.293 ± 0.177	-1.6	-
	Nest incubation	-0.329 ± 0.053	-6.2	<0.001
	Offspring provisioning	0.459 ± 0.057	8	<0.001
	Nest daily predation rate	1.060 ± 1.059	1.0	0.32
Research effort	-0.0001 ± 0.00007	-1.5	0.14	

Furthermore, we tried to make use of another variable related to the daily nest predation rate, i.e., the number of predator species sympatric with each species across its geographic range (8083 species), for the analysis. The data was obtained from Valcu et al. (2014). Although this variable does not directly reflect the intensity of nest predation, it provides us with the largest sample size for the analysis. After accounting for phylogenetic effects and matching with our dataset on sex-specific parental care, we had 1247 species. Using this data, we run a similar Model 4* (see below), where we used the number of predators instead of nest daily predation rate as a fixed effect.

		Estimate ($\beta \pm SE$)	t	p
Model 4*	Random effects:			
	Phylogeny (λ)	0.66 (0.57 – 0.73)		
	Fixed effects:			
	Intercept	0.109 ± 0.164	0.6	-
	Nest incubation	-0.189 ± 0.026	-7.4	<0.001
	Offspring provisioning	0.239 ± 0.025	9.4	<0.001
	Number of predators	0.003 ± 0.005	0.7	0.48
Research effort	-0.0002 ± 0.00004	-0.4	0.66	

The results of this Model 4* (now included in the Supplementary Information as Table S8) showed the same pattern as we found before, namely, there was no evidence of a significant association between predation risk and sex-specific parental care investment in birds.

References:

Matysioková, Beata, and Vladimír Remeš. "Evolution of parental activity at the nest is shaped by the risk of nest predation and ambient temperature across bird species." *Evolution* 72.10 (2018): 2214-2224.

Unzeta, Mar, Thomas E. Martin, and Daniel Sol. "Daily nest predation rates decrease with body size in passerine birds." *The American Naturalist* 196.6 (2020): 743-754.

Freeman, Benjamin G., et al. "Adaptation and latitudinal gradients in species interactions: nest predation in birds." *The American Naturalist* 196.6 (2020): E160-E166.

Valcu, Mihai, et al. "Global gradients of avian longevity support the classic evolutionary theory of ageing." *Ecography* 37.10 (2014): 930-938.

2. The strength of the paper is exploring the associations among care components and linking these to other components of sex roles, eg mating systems (see comments by Rev 1). Whilst the novelty of the work is in the large dataset and the clear presentation of the work, previous studies that explored similar associations should be credited (eg Long et al. 2022, Gonzalez-Voyer et al. 2022).

Thanks for acknowledging our efforts in compiling a large dataset that can be used to test the associations among care components (both phenotypically and phylogenetically, Figure 3) and link stage-specific parental

care with other components of sex roles. As illustrated above, the new papers you mentioned were not yet online when we submitted our manuscript. Now it is a good opportunity to incorporate them into the revision. We have now discussed both papers in the revised manuscript:

Lines 286-288, citing Gonzalez-Voyer et al. (2022): It contrasts with a recent phylogenetic analysis of the sex roles in birds, which found an increase of male-biased care as the proportion of broods with extra-pair offspring increases, albeit with a relatively small effect size⁷⁰.

Lines 309-311, citing Long et al. (2022): However, it concurs with a recent study which showed that avian species with open nests (considered as providing less protection from predators than closed nests) do not exhibit higher parental cooperation than species with closed nests⁸⁰.

In my view, the ecological and life-history predictors used in the MS are quite simplistic and given the huge complexity of both types of variables (ie ecology and life history), using a limited set of proxies for these can only uncover preliminary associations.

Thank you for your comment. We agree that the predictors (i.e., sexual selection, the certainty of paternity, nest predation risk, and offspring life history traits) used in this MS are limited. The four potential driving forces of parental care evolution were chosen because there are clear theoretical predictions of their effects (detailed explanations please paragraph 4 of the Introduction). We also agree that many more ecological and life history variables can potentially play a role in the evolution of stage-specific parental care patterns.

However, caution must be exercised when incorporating diverse variables into statistical models. Before including one potential predictor, we need to formulate a clear hypothesis related to it. Otherwise, the statistics are prone to suffer from multiple tests (too many predictors resulting in significant results randomly) and a HACK issue (including all potential predictors available and hypothesizing after the results are known) in ecology and evolution studies (Forstmeier et al. 2017). Considering the tradeoff between incorporating relevant biological factors and preserving statistical rigour and transparency, we believe the four major predictors and interactions (each testing a clear hypothesis) that we used in this study have achieved a good balance.

Nevertheless, we fully agree that more ecological and life history factors should be considered in future studies, particularly those that aim to understand exceptions to the common patterns. We highlight this and encourage future investigations in the last paragraph of the discussion.

Thanks again for your constructive critics. We carefully considered all of them and did our best to address each point. We hope you'll find our improvements satisfactory.

Reference

Forstmeier, Wolfgang, Eric-Jan Wagenmakers, and Timothy H. Parker. "Detecting and avoiding likely false-positive findings—a practical guide." *Biological Reviews* 92.4 (2017): 1941-1968.

REVIEWER COMMENTS

Reviewer #4 (Remarks to the Author):

I appreciate all the good efforts the authors put in revising the MS, and glad to see the main results remained robust after validating the scoring using independent sets of data (but see below). As a result, the MS has improved.

However, I still see a number of issues that makes the MS somehow inaccurate, difficult to follow and unconvincing at places.

1. The Abstract finishes somehow abruptly. What's the main take home messages of the MS? Why are the main findings novel and significant?

2. The Introduction treats some of the putative effects in a simplistic way. For instance, theoretical models show that the impact of paternity on care can be positive, negative or non-significant, see Westneat & Stewart 2003 *Ann Rev Ecol Evol & Syst* 34:365-396.

Also, the prediction that clutch size and/or offspring development predict trade-off between current and future RS and/or "reproductive value" is just very naive.

In short, the predictions and their justifications need lot more thinking and more nuanced presentation.

3. The Methods explains well the response variables, although for much of the main text, the two response vars are mixed up and difficult to follow.

First, the "sex-specific care pattern" is not really a good term since it is not sex-specific: it combines the care types of BOTH males and females. Perhaps "sex-difference in care" would be a better term, or simply "care pattern".

Note that "cooperation" is confusing from this point of view, since it is not clear whether it is a fourth type of care or cooperation is simply correspond to biparental care as far as Sex-specific care pattern concerned.

The term "form of care" is also not the best. Just by the name, most ornithologist would think "feeding the young", "defending the nest", "brooding the chicks" whereas here these 3 terms refer to different stages of reproduction: before offspring care commences, between egg completion and hatch, and between hatch and fledging/independence.

A confusion is that "incubation" refers both to a specific behaviour, eg keeping the eggs in a range of temperature, and also a stage in reproduction, ie egg-care.

I need to be clear that I'm fine with any sensible terminology, although experience taught me that the self-understanding and logical terms tend to be better used & cited.

4. Some of the main findings are in Table 1, although this table is pretty difficult to grasp. What are the response vars in each model? The response vars needs to be stated clearly in the Table legend.

Also, sample sizes presumably vary between models - so N should be stated clearly for all models in Table 1.

5. Comparative works are correlational in nature, so terms like "influence" or "affect" will need to be replaced by a more neutral terminology.

6. The Discussion is quite shallow. For instance, lines 388-391 are nothing else but piling up a number of questions without a logical structure nor it suggest any forward-looking newish idea.

7. The Methods do not provide the essential information on data validation. Data sources are missing and sample sizes are not provided (see lines 405-418). Data validation - as explained in the authors' Responses - is an important aspect of the study, and details should be provided in full in the Methods.

8. References need to be tidy up, eg Royle et al. 2012 is wrongly cited (ref 2).

Reviewer #4 (Remarks to the Author):

I appreciate all the good efforts the authors put in revising the MS, and glad to see the main results remained robust after validating the scoring using independent sets of data (but see below). As a result, the MS has improved.

However, I still see a number of issues that makes the MS somehow inaccurate, difficult to follow and unconvincing at places.

Thanks for acknowledging our efforts in improving the manuscript. We are grateful for the review and suggestions. We have carefully revised the manuscript in response to the reviewers' comments and believe that it is much improved. Please see our point-by-point reply below. Please note that in this reply, all line information refers to line numbers in the tracked version (main text file).

1. The Abstract finishes somehow abruptly. What's the main take home messages of the MS? Why are the main findings novel and significant?

We have thoroughly rewritten the Abstract and highlighted the take-home messages and the importance of our findings (lines 4-13).

2. The Introduction treats some of the putative effects in a simplistic way. For instance, theoretical models show that the impact of paternity on care can be positive, negative or non-significant, see Westneat & Stewart 2003 *Ann Rev Ecol Evol & Syst* 34:365-396.

We agree with the reviewer that the impact of paternity care can be of multiple dimensions. We have expanded our introduction on this. The introduction on the effects of paternity on parental care patterns has become more comprehensive (Lines 76–79):

“Although intuition suggests that the difference between male and female parents in expected parentage (e.g., due to female extra-pair mating) should produce female-biased care, theoretical models showed that the impact of paternity on care could be positive, negative, or non-significant^{12–14,42}.”

Also, the prediction that clutch size and/or offspring development predict trade-off between current and future RS and/or "reproductive value" is just very naive. In short, the predictions and their justifications need lot more thinking and more nuanced presentation.

We have carefully modified the description of the link between offspring's life history and parental investment by adding two additional modelling studies (Tamaru and Hōrak, 1999, Westneat and Mutzel, 2019) that showed more nuanced predictions (Lines 84–88):

“Because parents' caring efforts are linked to the trade-off between their current and future reproductive fitness, they are often expected to invest more in broods of higher reproductive value, e.g., broods of (artificially) enlarged sizes^{10,11,44–46}. However, theoretical models and experiments showed that parental investment may not always increase with the brood's reproductive value, with the adaptive behaviour sensitively dependent on environmental factors such as food supply^{47,48}.”

References:

Tamaru, Toomas, and Peeter Hōrak. "Should one invest more in larger broods? Not necessarily." *Oikos* (1999): 574-581.

Westneat, David F., and Ariane Mutzel. "Variable parental responses to changes in offspring demand have implications for life history theory." *Behavioral Ecology and Sociobiology* 73 (2019): 1-13.

3. The Methods explains well the response variables, although for much of the main text, the two response vars are mixed up and difficult to follow. First, the "sex-specific care pattern" is not really a good term since it is not sex-specific: it combines the care types of BOTH males and females. Perhaps "sex-difference in care" would be a better term, or simply "care pattern".

Thanks for your suggestion. We have changed "sex-specific care pattern" to either "sex-difference in care" or simply "care pattern" throughout the manuscript when appropriate. Please see the revised manuscript with marked changes for details.

Note that "cooperation" is confusing from this point of view, since it is not clear whether it is a fourth type of care or cooperation simply correspond to biparental care as far as Sex-specific care pattern concerned.

This is a fair point. We have changed "Cooperation" to "Cooperative Breeding" to avoid possible confusion with "Biparental care" throughout the text when appropriate.

The term "form of care" is also not the best. Just by the name, most ornithologist would think "feeding the young", "defending the nest", "brooding the chicks" whereas here these 3 terms refer to different stages of reproduction: before offspring care commences, between egg completion and hatch, and between hatch and fledging/independence.

Indeed, the three "forms of care" (i.e., nest building, incubation, and offspring provision) we focused on in this study occur at different stages of reproduction. However, we chose to use the term "form of care" instead of "stages of reproduction" because there are **other forms of care** that parents may provide during the corresponding reproductive stages. For example, parents may also clean the eggs and defend the nest after "nest building". Therefore, to avoid over-generalization, we used the term "form of care" rather than "stages of reproduction". We have clarified the terminology by defining the three care forms more clearly (Lines 377–378):

"The three forms (i.e., behaviours of parental care) were chosen due to their representativeness of distinct reproductive stages and the availability of well-documented data across species."

A confusion is that "incubation" refers both to a specific behaviour, eg keeping the eggs in a range of temperature, and also a stage in reproduction, ie egg-care. I need to be clear that I'm fine with any sensible terminology, although experience taught me that the self-understanding and logical terms tend to be better used & cited.

Related to our reply to the previous point, we have now defined "incubation" more clearly so that it refers solely to the specific behaviour of parental care, rather than the stage in reproduction when it occurs (Line 377-378).

4. Some of the main findings are in Table 1, although this table is pretty difficult to grasp. What are the response vars in each model? The response vars needs to be stated clearly in the Table legend. Also, sample sizes presumably vary between models – so N should be stated clearly for all models in Table 1.

Thanks for this constructive suggestion. We used two different ways to recode the response variables. Model 1 to Model 4 used the first way of recoding ('Female care', 'Biparental care' and 'Male care' as '-1', '0', and '+1', respectively; species in the 'Cooperative Breeding' category were also coded as '0', because breeders and helpers of both sexes contributed to care). Model 5 used the second way of recoding ('Female care' and 'Male care' as '1', 'Biparental care' as '2', and 'Cooperative Breeding' as '3'). We have now added statements of the response variables in the Table legend (Lines 137–141). We have also stated the sample size (N) clearly for all models in Table 1.

5. Comparative works are correlational in nature, so terms like “influence” or “affect” will need to be replaced by a more neutral terminology.

We have carefully proofread the manuscript and made changes when applicable. Please see the revised manuscript with marked changes for details.

6. The Discussion is quite shallow. For instance, lines 388–391 are nothing else but piling up a number of questions without a logical structure nor do they suggest any forward-looking new idea.

We would like to respectfully point out that lines 388–391 were not in the *Discussion* section of our last submission of the manuscript, but in the *Methods* section. Furthermore, we did not raise questions there (please see the screenshot below).

388 In particular, for precocial species in which the young are relatively mature and mobile from hatching (i.e.,
389 the young leaves the nest shortly after hatching), although parents usually do not feed the precocial chicks,
390 they still invest intensive care efforts (e.g., leading chicks to the food) until their offspring’s independence.
391 In this case, we classified the provisioner sex as the sex who cared for chicks before independence.

Nevertheless, we have carefully checked the *Discussion* of the revised manuscript to make sure that the questions we raised are relevant for future research and follow a clear logic structure, such as in Lines 243–246.

7. The Methods do not provide the essential information on data validation. Data sources are missing and sample sizes are not provided (see lines 405–418). Data validation - as explained in the authors' Responses - is an important aspect of the study, and details should be provided in full in the Methods.

We have elaborated on data validation by adding a section “*Cross-validation with independent datasets*” in the Methods (Lines 428–447). We also commented on the cross-validation of datasets compiled between two independent observers in the study (Lines 417–419, Lines 424–426).

Data sources are now explicitly provided in the text (e.g., BirdTree.org) as well as in the references. We also paid attention to include sample sizes wherever applicable in the revised text (Lines 483–494).

8. References need to be tidied up, e.g. Royle et al. 2012 is wrongly cited (ref 2).

We have checked the references carefully and believe that they are now correct.